# The large GTPase Sey1/atlastin mediates lipid droplet- and FadL-dependent intracellular fatty acid metabolism of *Legionella pneumophila*

Dario Hüsler[1], Pia Stauffer[1], Bernhard Keller[1], Desirée Böck[2†], Thomas Steiner[3], Anne Ostrzinski[4], Simone Vormittag[1], Bianca Striednig[1], A Leoni Swart[1‡], François Letourneur[5§], Sandra Maaß[4], Dörte Becher[4], Wolfgang Eisenreich[3], Martin Pilhofer[2], Hubert Hilbi[1]*

[1]Institute of Medical Microbiology, University of Zürich, Zürich, Switzerland; [2]Institute of Molecular Biology and Biophysics, ETH Zürich, Zürich, Switzerland; [3]Bavarian NMR Center - Structural Membrane Biochemistry, School of Natural Sciences, Technical University of Munich, Garching, Germany; [4]Institute of Microbiology, University of Greifswald, Greifswald, Germany; [5]UMR5294, LPHI, CNRS, INSERM, University of Montpellier, Montpellier, France

**\*For correspondence:**
hilbi@imm.uzh.ch

**Present address:** †Institute of Pharmacology and Toxicology, University of Zürich, Zürich, Switzerland; ‡Biozentrum, University of Basel, Basel, Switzerland; §VBIC, INSERM U1047, University of Montpellier, Montpellier, France

**Competing interest:** The authors declare that no competing interests exist.

**Abstract** The amoeba-resistant bacterium *Legionella pneumophila* causes Legionnaires' disease and employs a type IV secretion system (T4SS) to replicate in the unique, ER-associated *Legionella*-containing vacuole (LCV). The large fusion GTPase Sey1/atlastin is implicated in ER dynamics, ER-derived lipid droplet (LD) formation, and LCV maturation. Here, we employ cryo-electron tomography, confocal microscopy, proteomics, and isotopologue profiling to analyze LCV-LD interactions in the genetically tractable amoeba *Dictyostelium discoideum*. Dually fluorescence-labeled *D. discoideum* producing LCV and LD markers revealed that Sey1 as well as the *L. pneumophila* T4SS and the Ran GTPase activator LegG1 promote LCV-LD interactions. In vitro reconstitution using purified LCVs and LDs from parental or Δ*sey1* mutant *D. discoideum* indicated that Sey1 and GTP promote this process. Sey1 and the *L. pneumophila* fatty acid transporter FadL were implicated in palmitate catabolism and palmitate-dependent intracellular growth. Taken together, our results reveal that Sey1 and LegG1 mediate LD- and FadL-dependent fatty acid metabolism of intracellular *L. pneumophila*.

## Editor's evaluation

This important study advances our understanding of host-derived lipid droplets (LDs) interaction with intracellular pathogens. The use of amoeba species *Dictyostelium* discoideum as a host for *Legionella pneumophila* infection is compelling and goes beyond the current state of the art. The data were collected and analyzed using convincing methodology and this paper will interest cell biologists and microbiologists working on the interaction of microbes with host cells.

## Introduction

The causative agent of Legionnaires' disease, *Legionella pneumophila*, is a facultative intracellular bacterium, which adopts a similar mechanism to replicate in free-living protozoa and lung macrophages (*Newton et al., 2010*; *Boamah et al., 2017*; *Mondino et al., 2020*). To govern the interactions with

eukaryotic host cells, *Legionella* spp. employ the genus-conserved Icm/Dot type IV secretion system (T4SS), which in *L. pneumophila* translocates more than 300 different 'effector' proteins (*Qiu and Luo, 2017*; *Hilbi and Buchrieser, 2022*; *Lockwood et al., 2022*). The effector proteins subvert pivotal processes and establish a unique replication niche, the *Legionella*-containing vacuole (LCV), which communicates with the endosomal, secretory and retrograde vesicle trafficking pathways, but restricts fusion with lysosomes (*Isberg et al., 2009*; *Asrat et al., 2014*; *Finsel and Hilbi, 2015*; *Personnic et al., 2016*; *Sherwood and Roy, 2016*; *Bärlocher et al., 2017*; *Steiner et al., 2018a*; *Swart and Hilbi, 2020b*). Given that many cellular pathways and effector protein targets are conserved, the genetically tractable amoeba *Dictyostelium discoideum* is a versatile and powerful model to analyze pathogen-phagocyte interactions (*Cardenal-Muñoz et al., 2017*; *Swart et al., 2018*).

During LCV formation, the phosphoinositide (PI) lipid phosphatidylinositol 3-phosphate (PtdIns(3)*P*) is converted to PtdIns(4)*P* (*Weber et al., 2006*; *Weber et al., 2014b*; *Steiner et al., 2018a*; *Weber et al., 2018*; *Swart and Hilbi, 2020b*). Additionally, the LCV intercepts and fuses with endoplasmic reticulum (ER)-derived vesicles (*Kagan and Roy, 2002*; *Robinson and Roy, 2006*; *Arasaki et al., 2012*), and the LCV itself tightly associates with the ER (*Swanson and Isberg, 1995*; *Robinson and Roy, 2006*). The LCV-ER association persists even upon isolation and purification of intact LCVs (*Urwyler et al., 2009*; *Hoffmann et al., 2014*; *Schmölders et al., 2017*). The tight LCV-ER association has recently been confirmed by showing that LCVs form extended membrane contact sites (MCS) with the ER (*Vormittag et al., 2023*).

The ER is a highly dynamic organelle (*Shibata et al., 2006*; *Hu and Rapoport, 2016*; *Nixon-Abell et al., 2016*), and its morphology and dynamics are largely controlled by the reticulon family of membrane tubule-forming proteins (*Voeltz et al., 2006*; *Hu et al., 2008*) and the atlastin family of trans-membrane large fusion GTPases (*Hu et al., 2009*; *Orso et al., 2009*). Atlastins are conserved from yeast to plants and mammals (*Anwar et al., 2012*; *Zhang et al., 2013*), mediate the homotypic fusion of ER tubules and share a similar domain organization, which comprises an N-terminal GTPase domain linked through a helical bundle (HB) domain to two adjacent transmembrane segments and a C-terminal tail that contains an amphipathic helix (*Hu and Rapoport, 2016*). Structural and biochemical studies revealed that upon GTP binding, the GTPase and HB domains of atlastins on two distinct ER tubules dimerize, and the trans-homodimers pull together opposing membranes thus facilitating their fusion (*Bian et al., 2011*; *Byrnes and Sondermann, 2011*; *Liu et al., 2012*; *Byrnes et al., 2013*; *Liu et al., 2015*).

*D. discoideum* produces a single orthologue of human atlastin-1–3 (Atl1-3) termed Sey1, which shares the same domain organization as the mammalian atlastins (*Steiner et al., 2017*). Initially, Sey1, Atl3, and reticulon-4 (Rtn4) were identified by proteomics in intact LCVs purified from *L. pneumophila*-infected *D. discoideum* or macrophages (*Hoffmann et al., 2014*), and the localization of Sey1/Atl3 and Rtn4 to the ER surrounding LCVs was validated by fluorescence microscopy (*Haenssler et al., 2015*; *Steiner et al., 2017*). Sey1 is not implicated in the formation of the PtdIns(4)*P*-positive LCV membrane and not essential for the recruitment of ER, but promotes pathogen vacuole expansion and enhances intracellular replication of *L. pneumophila* (*Steiner et al., 2017*; *Steiner et al., 2018b*). The production of a catalytically inactive, dominant-negative Sey1_K154A mutant protein, or the depletion of mammalian Atl3, restricts *L. pneumophila* replication and impairs LCV maturation. *D. discoideum* Δ*sey1* mutant amoeba are enlarged but grow and develop similarly to the parental strain (*Hüsler et al., 2021*). The mutant strain shows pleiotropic defects, including aberrant ER architecture and dynamics, inability to cope with prolonged ER stress, defective intracellular proteolysis, cell motility and growth on bacterial lawns (*Hüsler et al., 2021*). In the Δ*sey1* mutant amoeba LCV-ER interactions, LCV expansion and intracellular *L. pneumophila* replication are impaired, similar to what was observed with *D. discoideum* producing dominant negative Sey1 (*Steiner et al., 2017*). Taken together, Sey1/Atl3 controls circumferential ER remodeling during LCV maturation and intracellular replication of *L. pneumophila* (*Steiner et al., 2017*; *Steiner et al., 2018b*; *Hüsler et al., 2021*).

In addition to promoting ER dynamics, atlastins contribute to a number of other cellular processes including the biogenesis of ER-derived lipid droplets (LDs; *Klemm et al., 2013*). LDs are the major cellular storage compartments of neutral lipids; however, they are also involved in many other cellular processes such as energy homeostasis, lipid metabolism, generation of membrane lipids and signaling molecules as well as retention of harmful proteins and lipids (*Walther and Farese, 2012*; *Hashemi and Goodman, 2015*; *Welte, 2015*; *Kimmel and Sztalryd, 2016*; *Welte and Gould, 2017*). In *D.*

*discoideum*, LDs accumulate upon feeding the cells with fatty acids (in particular palmitate) or bacteria (*Du et al., 2013*). Proteomics and lipidomics analysis of *D. discoideum* LDs revealed that the lipid constituents are similar to mammalian LDs and comprise mainly triacylglycerol (57%), free fatty acids (22 %) and sterol esters (4%). LDs are coated by a polar phospholipid monolayer and distinct proteins (*Du et al., 2013*), such as perilipin (*Miura et al., 2002*), and small GTPases, as well as ER proteins (reticulon C, RtnlC; protein disulfide isomerase, PDI; lipid droplet membrane protein, LdpA; and 15 lipid metabolism enzymes), the latter reflecting their cellular organelle origin (*Du et al., 2013*). LDs are transported along microtubules and actin filaments or moved by actin polymerization (*Welte, 2004*; *Welte, 2009*; *Pfisterer et al., 2017*; *Welte and Gould, 2017*; *Kilwein and Welte, 2019*), and they form contact sites with various cell organelles (*Kumar et al., 2018*; *Benador et al., 2019*; *Yeshaw et al., 2019*; *Herker et al., 2021*).

Only after the replication-permissive LCV has been formed, *L. pneumophila* engages in intracellular replication. The bacteria employ a biphasic lifestyle comprising a transmissive (motile, virulent) and a replicative phase (*Molofsky and Swanson, 2004*). *L. pneumophila* is an obligate aerobe bacterium, which previously has been thought to rely on certain amino acids as carbon and energy source (*Abu Kwaik and Bumann, 2013*; *Manske and Hilbi, 2014*). Indeed, isotopologue profiling studies with stable $^{13}$C-isotopes indicated that serine is a major carbon and energy source for *L. pneumophila* and readily metabolized by the bacteria (*Eylert et al., 2010*). More recent physiological and isotopologue profiling studies established that glucose, inositol, and glycerol are also metabolized by *L. pneumophila* under extracellular and intracellular conditions (*Eylert et al., 2010*; *Harada et al., 2010*; *Häuslein et al., 2016*; *Manske et al., 2016*). Finally, isotopologue profiling studies indicated that extracellular *L. pneumophila* efficiently catabolizes exogenous [1,2,3,4-$^{13}$C$_4$]palmitic acid, yielding $^{13}$C$_2$-acetyl-CoA, which is used to synthesize the storage compound polyhydroxybutyrate (PHB; *Häuslein et al., 2017*).

It is unknown how fatty acids are taken up by *L. pneumophila*. In *E. coli*, the long-chain fatty acid transporter FadL localizes to the outer membrane, where the monomeric protein adopts a 14-stranded, anti-parallel β barrel structure (*van den Berg et al., 2004*). The N-terminal 42 amino acid residues of FadL form a small 'hatch' domain that plugs the barrel, and the hydrophobic substrate leaves the transporter by lateral diffusion into the outer membrane (*Hearn et al., 2009*). *L. pneumophila* encodes a homolog of *E. coli* FadL, Lpg1810, which was identified as a surface-associated protein by fluorescence-labeling and subsequent mass spectrometry (MS), confirming its presence in the bacterial outer membrane (*Khemiri et al., 2008*).

Given the role of LDs as lipid storage organelles regulated by atlastins, we set out to analyze the contribution of LDs, Sey1 and FadL for intracellular replication and palmitate catabolism of *L. pneumophila* in *D. discoideum*. We found that Sey1 regulates LD protein composition and promotes Icm/Dot-dependent LCV-LD interactions as well as FadL-dependent fatty acid metabolism of intracellular *L. pneumophila*.

## Results

### Palmitate-induced LDs interact with LCVs in *D. discoideum*

To initially explore whether fatty acids and/or LDs play a role for intracellular replication of *L. pneumophila*, we fed *D. discoideum* strain Ax3 with palmitate, and assessed intracellular replication of the *L. pneumophila* wild-type strain JR32 or the mutant strain Δ*icmT*, which lacks a functional T4SS and is defective for effector protein secretion. Feeding with 200 μM palmitate overnight significantly promoted the intracellular growth of *L. pneumophila*, while higher concentrations of palmitate had a negative effect on growth (*Figure 1A*). This growth reduction was not owing to fatty acid toxicity for *D. discoideum*, as up to 800 μM palmitate were not toxic for the amoeba (*Figure 1—figure supplement 1*).

To assess whether the growth-promoting effect of palmitate might involve LDs, we thought to visualize the possible interactions between LCVs and LDs. To this end, *D. discoideum* strain Ax3 was fed with palmitate, infected with the *L. pneumophila* wild-type strain JR32 and subjected to cryo-electron tomography (cryoET). The obtained cryotomograms clearly show an intensive interaction between LCVs and LDs (*Figure 1B*). Upon contact with the LCV, the LDs tightly interact with the limiting membrane of the pathogen vacuole and even appear to integrate into the LCV limiting membrane. We did not observe tethering of LDs to the LCV with the LD lipid monolayer and the LCV lipid bilayer

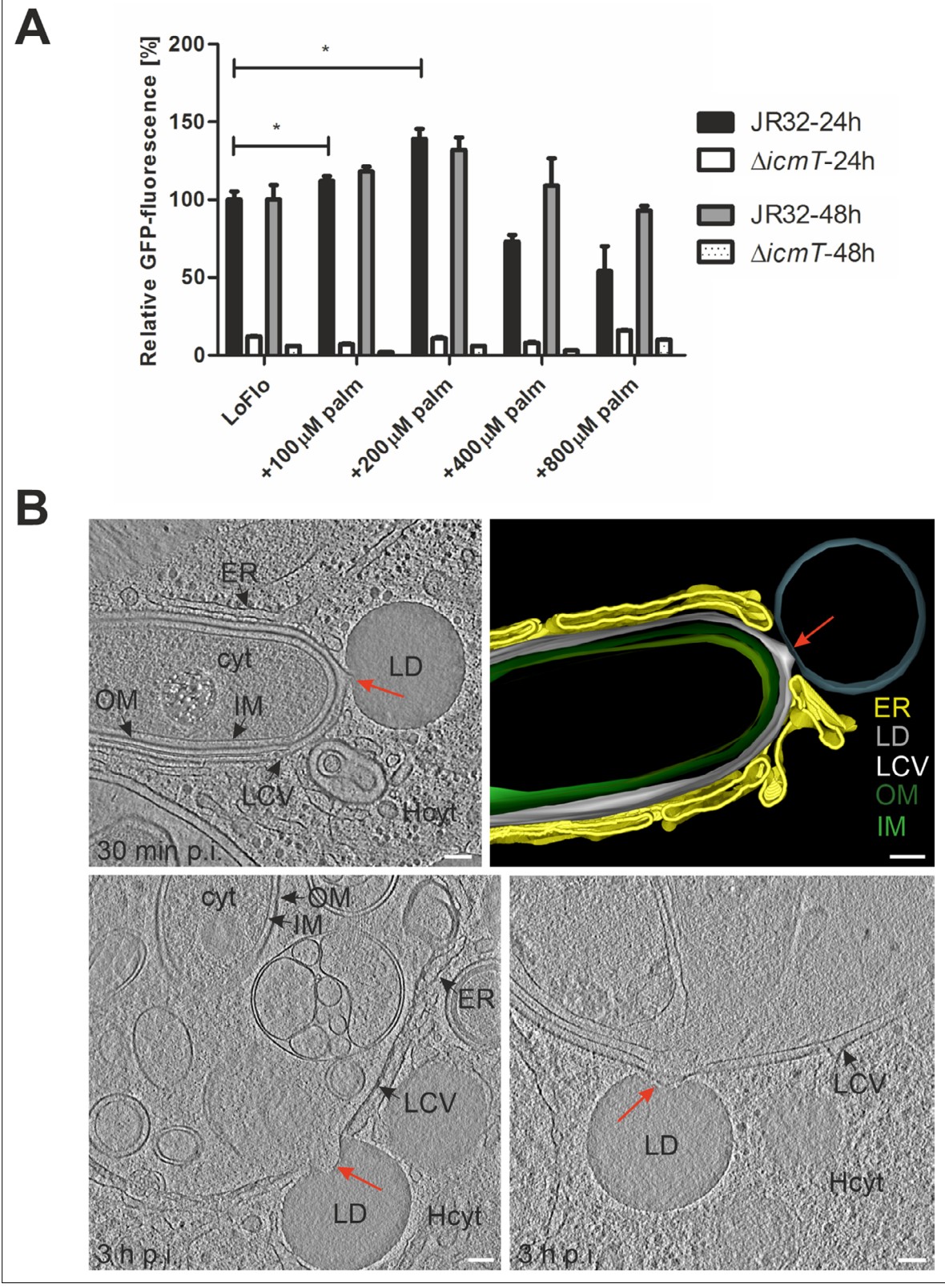

**Figure 1.** Palmitate-induced lipid droplets interact with LCVs in *D. discoideum*. (**A**) *D. discoideum* Ax3, untreated (LoFlo medium) or treated with increasing concentrations of sodium palmitate (100–800 µM, 3 hr), were infected (MOI 10) with GFP-producing *L. pneumophila* wild-type JR32 or Δ*icmT* (pNT28). The GFP-fluorescence was measured with a microtiter plate reader at 1 hr, 24 hr, and 48 hr p.i. Data show the relative fluorescence increase between 1 hr and 24 hr or 48 hr p.i. (JR32: black/grey bar; Δ*icmT*: white/dotted bar). Data represent means ± SD of three independent

*Figure 1 continued on next page*

*Figure 1 continued*

experiments (*p<0.05). (**B**) Representative cryotomograms of *D. discoideum* Ax3, fed (3 hr) with 200 µM sodium palmitate and infected (MOI 100) with *L. pneumophila* JR32 for 30 min (**top**) or 3 hr (**bottom**). Intimate LCV-LD interactions are clearly visible (red arrows). Reconstruction of LCV-LD interaction observed at 30 min p.i. (**top**; **right**). OM, outer membrane; IM, inner membrane; LCV, *Legionella*-containing vacuole (limiting membrane); ER, endoplasmic reticulum; LD, lipid droplet; cyt, *L. pneumophila* cytoplasm; Hcyt, host cell cytoplasm. Scale bars: 100 nm.

The online version of this article includes the following figure supplement(s) for figure 1:

**Figure supplement 1.** Cytotoxicity of palmitate.

spanning a discrete, short distance in the nm range. Hence, LDs undergo robust and intimate interactions with LCVs in infected *D. discoideum*.

## Sey1 promotes LD recruitment to intact LCVs in *D. discoideum*

Given that large GTPases of the atlastin family are implicated in LD formation in mammalian cells (*Klemm et al., 2013*), we next assessed whether Sey1 affects early LCV-LD interactions in *D. discoideum*. To this end, we used dually fluorescence-labelled *D. discoideum* producing the LCV marker P4C-GFP and the LD marker mCherry-Plin. The *D. discoideum* parental strain Ax3 or Δ*sey1* mutant amoeba were fed overnight with 200 µM palmitate, stained with LipidTOX Deep Red and infected with mCerulean-producing *L. pneumophila* JR32. Within the first hour of infection, the dynamic interactions of single LCVs with LDs were recorded for 60 s each at different time points (*Figure 2A*, *Figure 2—figure supplement 1*). As the LCVs matured over the course of 1 hr post infection (p.i.), the overall LCV-LD contact time gradually increased in *D. discoideum* Ax3, while it remained lower in Δ*sey1* mutant amoeba (*Figure 2B*). Moreover, the retention time of individual LDs on LCVs was also signficantly higher in strain Ax3 than in Δ*sey1* mutant amoeba (*Figure 2C*). Taken together, these real-time data indicate that Sey1 promotes the dynamics of LCV-LD interactions during the course of LCV maturation.

To assess the integrity of the LCVs during their interaction with LDs, we used *D. discoideum* Ax3 or Δ*sey1* producing P4C-GFP and cytoplasmic mCherry (*Figure 2D*). The production of cytoplasmic mCherry allows to assess the integrity of pathogen vacuoles in the course of *D. discoideum* infection (*Koliwer-Brandl et al., 2019*). The *D. discoideum* strains were fed overnight with 200 µM palmitate, infected with mCerulean-producing *L. pneumophila* JR32, and LDs were stained with LipidTOX Deep Red. This approach revealed that 1 hr post infection, all LCVs formed in either *D. discoideum* Ax3 or Δ*sey1* mutant amoeba were impermeable to cytoplasmic mCherry, and therefore, LCV membrane integrity was not compromised (*Figure 2D*). In addition, these experiments confirmed that LCVs in *D. discoideum* Ax3 are decorated with approximately twice as many LDs as LCVs in the Δ*sey1* strain (*Figure 2E*).

## The *L. pneumophila* T4SS promotes Sey1-dependent LCV-LD interactions

Next, we sought to validate that Sey1 promotes LCV-LD interactions in palmitate-fed, fixed *D. discoideum* and to test if the process also depends on the *L. pneumophila* Icm/Dot T4SS. To this end, we used *D. discoideum* producing mCherry-Plin and AmtA-GFP, a probe localizing to vacuoles containing either wild-type strain JR32 or Δ*icmT* mutant bacteria (*Figure 3A*). This approach indicated that the mean number of LDs localizing to LCVs harboring strain JR32 was more than twice as high in *D. discoideum* Ax3 as compared to the Δ*sey1* mutant amoeba, and the effect was of similar magnitude, when the number of LDs per LCV area was calculated (*Figure 3B*). Contrarily, Sey1 did not promote the interaction of vacuoles harboring Δ*icmT* mutant bacteria with LDs, and overall, significantly fewer LDs associated with these vacuoles (*Figure 3B*). Taken together, these studies using fixed *D. discoideum* amoeba reveal that Sey1 promotes LCV-LD interactions and the Icm/Dot T4SS is required for LD accumulation on LCVs.

Atlastins regulate the number and size of LDs in mammalian cells (*Klemm et al., 2013*). Hence, we assessed the role of Sey1 for the number and size of LDs in *D. discoideum*. *D. discoideum* producing the LD marker mCherry-Plin and the phagosome marker AmtA-GFP were left untreated or fed with 200 µM palmitate overnight, fixed and stained with LipidTOX Deep Red, and the number and size of LDs were quantified. *D. discoideum* Ax3 or Δ*sey1* amoeba were found to harbor approximately 10 LDs per cell, regardless of whether they were uninfected or infected with *L. pneumophila* wild-type JR32

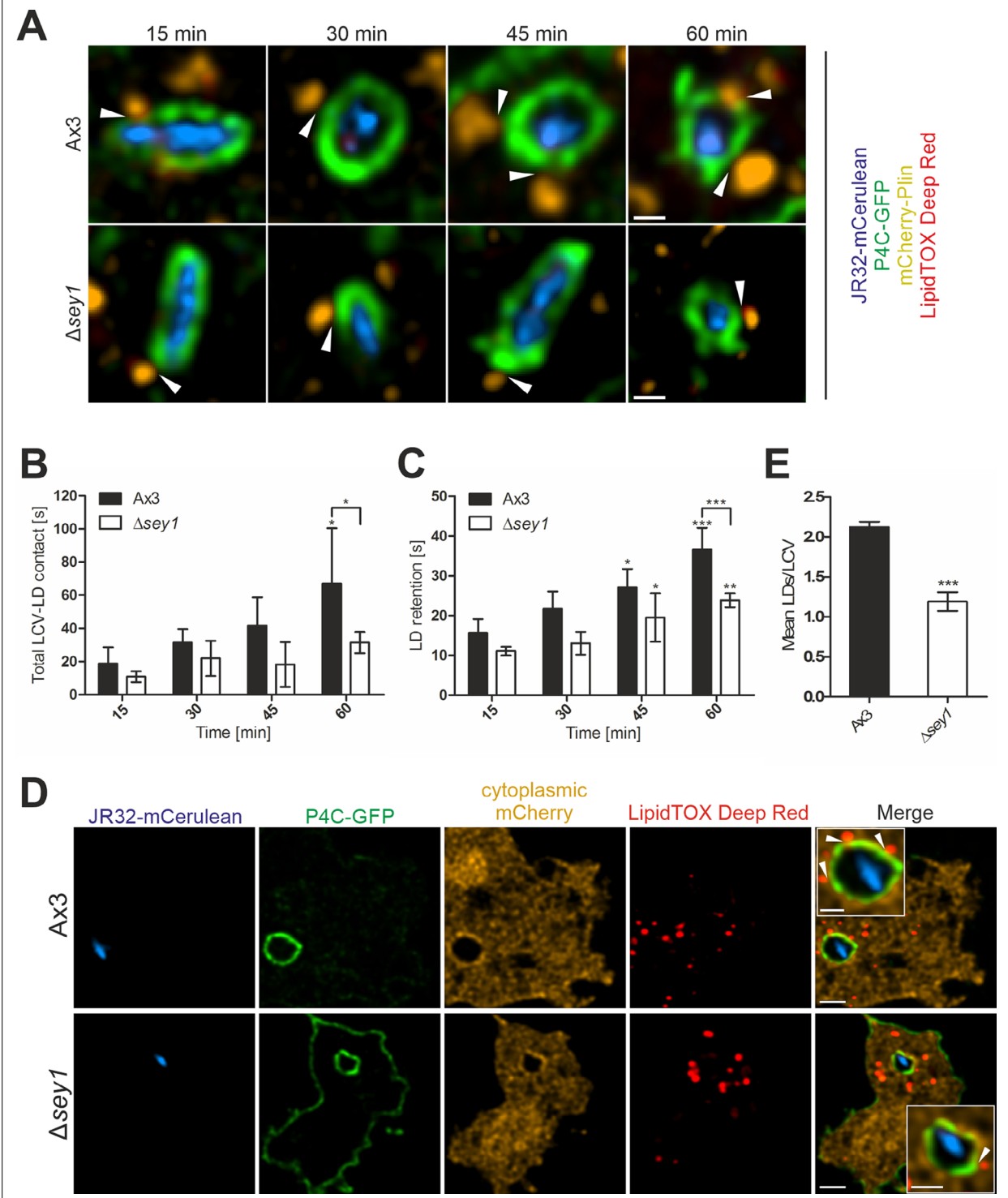

**Figure 2.** Sey1 promotes LDs recruitment to intact LCVs in *D. discoideum*. (**A**) Representative fluorescence micrographs of *D. discoideum* Ax3 or Δ*sey1* producing P4C-GFP (pWS034) and mCherry-Plin (pHK102), fed overnight with 200 µM sodium palmitate, stained with LipidTOX Deep Red and infected (MOI 5) with mCerulean-producing *L. pneumophila* JR32 (pNP99). Infected cells were recorded for 60 s each at the times indicated. Examples are shown for contact between LDs and the LCV membrane (**white arrowheads**). Scale bars: 0.5 µm. (**B**) Quantification of (**A**), total contact time of LDs with the LCV recorded for 60 s at the indicated time points p.i. ($n^{LCV\text{-}LD\ contacts} > 30$). Data represent means ± SD of three independent experiments (*p<0.05). (**C**) Quantification of (**A**), retention time of single LDs with the LCV recorded for 60 s at the indicated time points p.i. ($n^{LCV\text{-}LD\ contacts} > 30$). Data represent means ± SD of three independent experiments (*p<0.05, **p<0.01; ***p<0.001). (**D**) Representative fluorescence micrographs of *D. discoideum* Ax3 or Δ*sey1* producing P4C-GFP (pWS034) and cytosolic mCherry (pDM1042), fed overnight with 200 µM sodium palmitate and infected (MOI 10, 1 hr) with

*Figure 2 continued on next page*

*Figure 2 continued*

mCerulean-producing *L. pneumophila* JR32 (pNP99), fixed with PFA and stained with LipidTOX Deep Red. Examples are shown for contact between LDs and the LCV membrane (**white arrowheads**). Scale bars: overview (2 µm), inset (1 µm). (**E**) Quantification of (**D**), mean number of LDs contacting a single LCV ($n^{LCVs}$ >102). Data represent means ± SD of three independent experiments (\*\*\*p<0.001).

The online version of this article includes the following figure supplement(s) for figure 2:

**Figure supplement 1.** Sey1 promotes LD recruitment to LCVs in *D. discoideum*.

or Δ*icmT* (*Figure 3—figure supplement 1*). Palmitate feeding did not significantly change the size of the LDs. Upon feeding *D. discoideum* with 200 µM palmitate overnight, uninfected or Δ*icmT*-infected amoeba contained significantly more LDs (ca. eightfold), and amoeba infected with *L. pneumophila* JR32 contained only ca. fourfold more LDs (*Figure 3—figure supplement 1*). Taken together, feeding with palmitate increased the number but not the size of LDs in *D. discoideum*, and infection with wild-type *L. pneumophila* reduced the LDs number compared to uninfected or Δ*icmT*-infected amoeba. However, in apparent contrast to mammalian cells, Sey1 did not seem to affect the number and size of LDs in *D. discoideum*.

We also quantified the ratio of LDs per LCV and LDs per LCV area in *D. discoideum* Ax3 or Δ*sey1* mutant amoeba, which were unstimulated or fed with 200 µM palmitate overnight (*Figure 3—figure supplement 1*). In unstimulated as well as in palmitate-fed *D. discoideum*, the number of LDs per LCV and the number of LDs per LCV area was significantly larger in the parental *D. discoideum* strain Ax3 as compared to Δ*sey1* mutant amoeba. Therefore, feeding with palmitate does not affect the stimulation of LCV-LD interactions by Sey1. However, in agreement with an increased overall number of LDs per cell, the overall number of LDs per LCV or LDs per LCV area increased upon feeding the amoeba with palmitate (*Figure 3—figure supplement 1*). Taken together, these results indicate that while feeding *D. discoideum* with palmitate increases the total number of LDs in amoeba and on LCVs, palmitate feeding does not change the positive effect of Sey1 on LCV-LD interactions also seen in unstimulated amoeba.

## Proteomics analysis of purified LDs identifies RanA GTPase and RanBP1

To gain further insights into the possible role of Sey1 for LD composition, we performed a comparative proteomics analysis of LDs from *D. discoideum*. To this end, LDs were harvested from palmitate-fed mCherry-Plin-producing *D. discoideum* Ax3 or Δ*sey1* mutant amoeba, purified by sucrose gradient centrifugation (*Figure 4—figure supplement 1*), and subjected to tandem mass spectrometry. This approach revealed 144 differentially produced proteins ($\log_2$ fold change > |0.8|), including some enzymes implicated in lipid metabolism (phospholipase PldA, phosphatidylinositol phosphate kinase Pik6/PIPkinA, sterol methyl transferase SmtA, acetoacetyl-CoA hydrolase) (*Supplementary file 1*, *Figure 4—figure supplement 1*). Among the differentially produced proteins, 7 or 22 were exclusively detected in LDs isolated from strain Ax3 or Δ*sey1*, respectively. Sey1 was identified on LDs isolated from strain Ax3, but as expected not on LDs isolated from Δ*sey1* mutant amoeba. Contrarily, the phospholipase PldA, the ER protein calnexin (CnxA) and the protein SCFD1/SLY1 implicated in ER to Golgi transport were identified only on LDs isolated from the Δ*sey1* strain (*Supplementary file 1*). The 50 most highly abundant proteins, which were not significantly different on LDs isolated from Ax3 or Δ*sey1* mutant amoeba, included perilipin (Plin, PlnA), which is involved in the formation and maintenance of LDs (*Du et al., 2013*), as well as – to our surprise – the small GTPase RanA (*Du et al., 2013*) and its effector RanBP1 (*Supplementary file 1*). Intriguingly, RanA is activated in *L. pneumophila*-infected cells and implicated in microtubule stabilization and LCV motility (*Rothmeier et al., 2013*; *Swart et al., 2020c*).

To assess the localization of Sey1 with regard to LDs, we used *D. discoideum* Ax3 producing GFP-Sey1 as well as the LD marker mCherry-Plin, and further stained LDs with LipidTOX Deep Red (*Figure 4—figure supplement 1*). Under the conditions used, GFP-Sey1 accumulated in the vicinity of LDs in intact cells as well as in cell homogenates, but apparently did not co-localize with LDs. This staining pattern suggests that Sey1 localizes only in very low amounts to LDs, or that localization of Sey1 to LDs is impaired due to the fluorescent protein tag.

To assess the localization of RanA and RanBP1 to LDs, we used *D. discoideum* Ax3 producing either RanA-mCherry or RanBP1-GFP and GFP-Plin or mCherry-Plin and further stained LDs with LipidTOX Deep Red (*Figure 4—figure supplement 1*). Under these conditions, ectopically produced

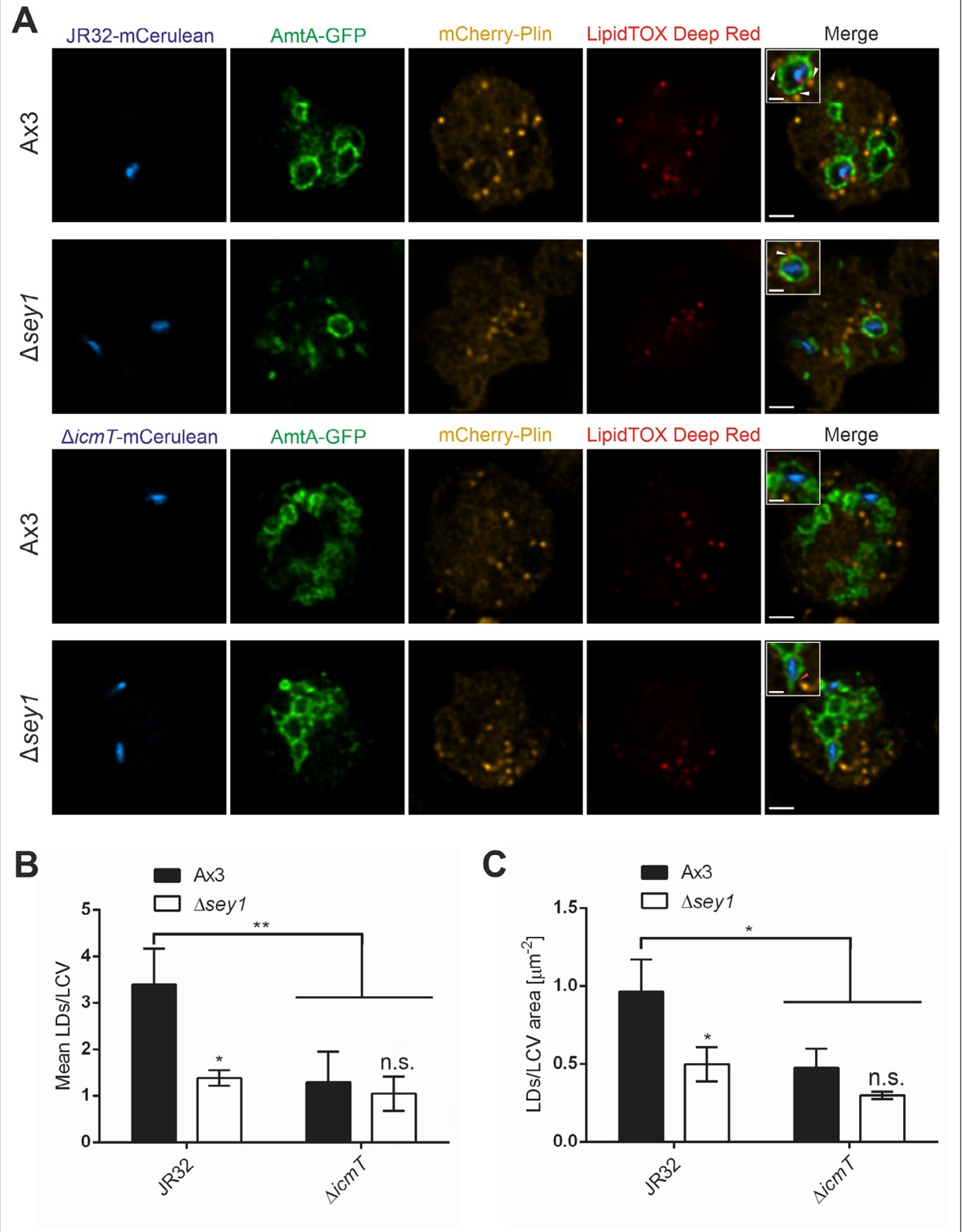

**Figure 3.** The *L. pneumophila* T4SS promotes Sey1-dependent LCV-LD interactions. (**A**) Representative fluorescence micrographs of *D. discoideum* Ax3 or Δ*sey1* producing AmtA-GFP (pHK121) and mCherry-Plin (pHK102), fed overnight with 200 μM sodium palmitate and infected (MOI 10, 1 hr) with mCerulean-producing *L. pneumophila* JR32 (**top**) or Δ*icmT* (**bottom**) (pNP99), fixed with PFA and stained with LipidTOX Deep Red. Examples are shown for contact between LDs and the LCV membrane (**white arrowheads**) or no contact (**red arrowhead**). Scale bars: overview (2 μm), inset (1 μm).

*Figure 3 continued on next page*

*Figure 3 continued*

(**B**) Quantification of (**A**), mean number of LDs contacting a single LCV (**left**) and ratio of LD number contacting one LCV divided by the LCV area (**right**) ($n^{LCVs}$ = 30). Data represent means ± SD of three independent experiments. (n.s., not significant; *p<0.05; **p<0.01).

The online version of this article includes the following figure supplement(s) for figure 3:

**Figure supplement 1.** Palmitate promotes lipid droplet biogenesis.

RanA-mCherry or RanBP1-GFP localized to membranous structures in the cell, including to Plin- and LipidTOX Deep Red-positive LDs. In summary, comparative proteomics of LDs isolated from *D. discoideum* Ax3 or *Δsey1* revealed that the phospholipase PldA is present exclusively in the mutant amoeba, and Plin, RanA as well as RanBP1 are detected in LDs from both *D. discoideum* strains.

## *L. pneumophila* LegG1 promotes Sey1-dependent LCV-LD interactions

Next, we sought to identify *L. pneumophila* effector proteins, which possibly determine LCV-LD interactions. The RCC1 repeat domain effector LegG1 activates the small GTPase RanA, which in its active, GTP-bound form interacts with RanBP1 and promotes microtubule stabilization (*Rothmeier et al., 2013*; *Swart et al., 2020c*). Since we found that LDs harbor RanA and RanBP1 (*Supplementary file 1*), we tested the hypothesis that LegG1 is implicated in LCV-LD dynamics. To this end, we infected palmitate-fed *D. discoideum* Ax3 or *Δsey1* producing P4C-GFP and mCherry-Plin with mCerulean-producing *L. pneumophila* JR32 or *ΔlegG1* and additionally stained LDs with LipidTOX Deep Red (*Figure 4A*, *Videos 1–4*). At 1 hr or 2 hr p.i., the overall LCV-LD contact time was lowered by ca. 50% upon infection with *ΔlegG1* (compared to JR32) or in *Δsey1* mutant *D. discoideum* (compared to strain Ax3) (*Figure 4B*). Intriguingly, the overall LCV-LD contact time was further significantly reduced upon infection of *Δsey1* mutant amoeba with *ΔlegG1* mutant bacteria (*Figure 4B*). Similar results were obtained by quantifying the retention time of individual LDs on LCVs (*Figure 4B*). The defects of the *ΔlegG1* mutant strain regarding the duration of LCV-LD contacts were complemented by providing the *legG1* gene on a plasmid (*Figure 4C*). In summary, these results indicate that the host large GTPase Sey1, as well as the *L. pneumophila* Ran GTPase activator LegG1 promote and additively affect the dynamics of LCV-LD interactions.

To assess the subcellular localization of the *L. pneumophila* effector LegG1, we used *D. discoideum* Ax3 or *Δsey1* producing GFP-LegG1 and P4C-mCherry, infected the amoeba with the *L. pneumophila* wild-type strain JR32 producing mCerulean, and further stained LDs with LipidTOX Deep Red (*Figure 4—figure supplement 1*). Ectopically produced GFP-LegG1 showed a punctate localization in the *D. discoideum* cytoplasm and also localized to LCVs in *D. discoideum Δsey1* as well as in Ax3, as previously published for M45-tagged LegG1 in strain Ax3 (*Rothmeier et al., 2013*). GFP-LegG1 also appeared to accumulate around LipidTOX Deep Red-positive LDs, in agreement with the notion that LegG1 accumulates in the vicinity of LDs.

The effector LegG1 promotes RanA activation, microtubule stabilization and LCV motility along microtubules (*Rothmeier et al., 2013*; *Swart et al., 2020c*). To test whether LegG1 affects microtubules in *D. discoideum*, we infected *D. discoideum* Ax3 or *Δsey1* producing GFP-tubulin A and P4C-mCherry with DsRed-producing *L. pneumophila* JR32, *ΔicmT*, *ΔlegG1* or *ΔlegG1*/pLegG1, and LDs in the infected amoeba were further stained with LipidTOX Deep Red (*Figure 4D*). Microtubule stability was impaired in *ΔlegG1*-infected *D. discoideum* Ax3, as published previously (*Rothmeier et al., 2013*), as well as in *D. discoideum Δsey1*. Accordingly, LegG1 promotes microtubule stability independently of Sey1. Microtubule stabilization was restored upon complementing the *ΔlegG1* mutant strain with plasmid-borne LegG1. In summary, these results indicate that LegG1 promotes LCV-LD interactions through the stabilization of microtubules. Moreover, the results indicate that the bacterial effector LegG1 (targeting RanA GTPase) and the eukaryotic large GTPase Sey1 (affecting the ER and LDs) function in different pathways to promote LCV-LD interactions.

## Sey1 and GTP promote LCV-LD interactions in vitro

Since Sey1 is a large fusion GTPase, we sought to test the nucleotide requirement of the LCV-LD interactions. To this end, we purified LCVs from *D. discoideum* Ax3 producing P4C-GFP infected with mCerulean-producing *L. pneumophila* JR32, mixed the pathogen vacuoles with purified LDs from palmitate-fed strain Ax3 producing mCherry-Plin and added 5 mM of different nucleotides

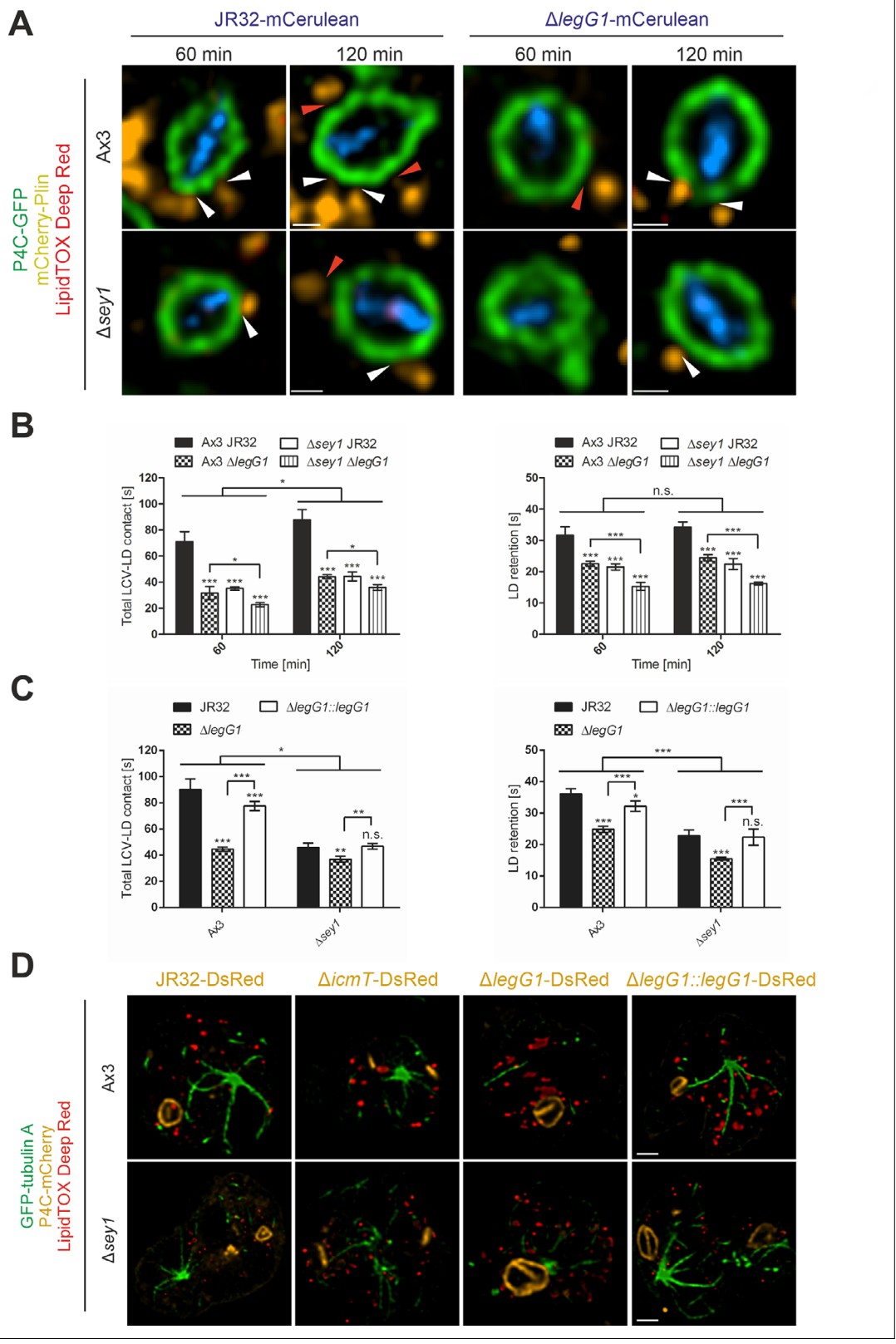

**Figure 4.** *L. pneumophila* LegG1 promotes Sey1-dependent LCV-LD interactions. (**A**) Representative fluorescence micrographs of *D. discoideum* Ax3 or Δ*sey1* producing P4C-GFP (pWS034) and mCherry-Plin (pHK102), fed overnight with 200 µM sodium palmitate, stained with LipidTOX Deep Red and infected (MOI 5) with *L. pneumophila* JR32 or Δ*legG1* producing mCerulean (pNP99). Infected cells were recorded for 60 s each at the

*Figure 4 continued on next page*

*Figure 4 continued*

times indicated. Examples are shown for contact between LDs and the LCV membrane (**white arrowheads**) or no contact (**red arrowheads**). Scale bars: 0.5 µm. (**B**) Quantification of (**A**), total contact time of LDs with the LCV (**left**) or retention time of single LDs with the LCV (**right**) recorded for 60 s at the indicated time points p.i. ($n^{\text{LCV-LD contacts}}$ > 30). Data represent means ± SD of three independent experiments (n.s., not significant; *p<0.05; ***p<0.001). (**C**) Quantification of total contact time of LDs with the LCV (**left**) or retention time of single LDs with the LCV (**right**) recorded for 60 s at 120 min p.i. ($n^{\text{LCV-LD contacts}}$ > 30). *D. discoideum* Ax3 or Δ*sey1* producing P4C-GFP (pWS034) and mCherry-Plin (pHK102), fed overnight with 200 µM sodium palmitate, stained with LipidTOX Deep Red and infected (MOI 5) with *L. pneumophila* JR32 or Δ*legG1* producing DsRed (pSW001), or Δ*legG1* producing DsRed and M45-LegG1 (pER005; Δ*legG1::legG1*) were analysed. Data represent means ± SD of three independent experiments (n.s., not significant; *p<0.05; **p<0.01; ***p<0.001). (**D**) Representative fluorescence micrographs of *D. discoideum* Ax3 or Δ*sey1* producing GFP-tubulin A (pLS110) and P4C-mCherry (pWS032), fed overnight with 200 µM sodium palmitate and infected (MOI 10, 1 hr) with *L. pneumophila* JR32, Δ*icmT* or Δ*legG1* producing DsRed (pSW001), or Δ*legG1* producing DsRed and M45-LegG1 (pER005; Δ*legG1::legG1*), fixed with PFA and stained with LipidTOX Deep Red. Scale bars: 2 µm.

The online version of this article includes the following figure supplement(s) for figure 4:

**Figure supplement 1.** Purification of LDs, proteomics and localization of Sey1, RanA, and RanBP1.

**Figure supplement 2.** Cellular localization of GFP-LegG1.

---

(***Figure 5A***). Using this in vitro reconstitution approach, the addition of GTP resulted in a ca. 2.5-fold higher number of LDs per LCV as compared to the addition of GDP, Gpp(NH)p or GTPγS (***Figure 5B***).

In an analogous approach, we tested whether Sey1 present in the purified LCV or LD fraction promotes the interaction between the two compartments. We mixed LCVs purified from *D. discoideum* Ax3 or Δ*sey1* producing P4C-GFP infected with mCerulean-producing *L. pneumophila* JR32 with purified LDs from palmitate-fed strain Ax3 or Δ*sey1* producing mCherry-Plin in presence of 5 mM GTP or GDP (***Figure 5C***). The mean number of LDs per LCV was highest for both compartments isolated from *D. discoideum* Ax3, followed by LDs purified from strain Ax3 and LCVs from Δ*sey1* mutant amoeba (***Figure 5D***). Contrarily, the LCV-LD interactions were impaired for Δ*sey1*-derived LDs, suggesting that Sey1 regulates LD traits implicated in the interactions between the two compartments. The addition of GDP to the reconstitution assays yielded only background levels of LD/LCV ratios. In summary,

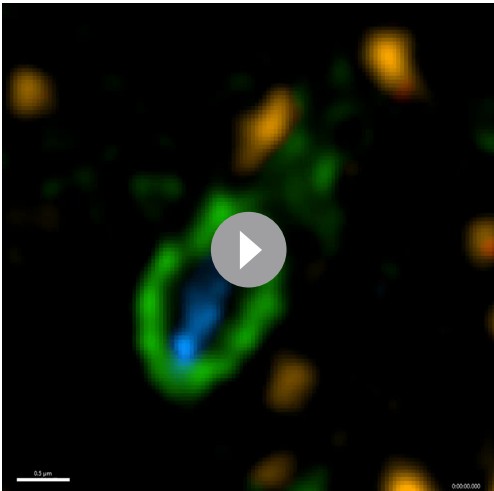

**Video 1.** *D. discoideum* Ax3 infected with *L. pneumophila* JR32. Representative movie of *D. discoideum* Ax3 producing P4C-GFP (pWS034) and mCherry-Plin (pHK102), fed overnight with 200 µM sodium palmitate, stained with LipidTOX Deep Red and infected (MOI 5) with mCerulean-producing *L. pneumophila* JR32 (pNP99). Infected cells were recorded for 60 s each at the times indicated.

https://elifesciences.org/articles/85142/figures#video1

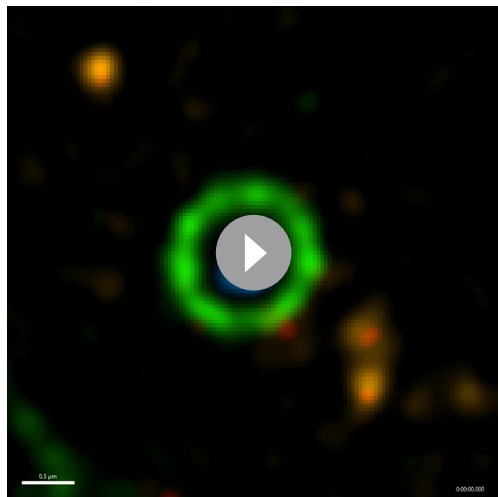

**Video 2.** *D. discoideum* Ax3 infected with *L. pneumophila* Δ*legG1*. Representative movie of *D. discoideum* Ax3 producing P4C-GFP (pWS034) and mCherry-Plin (pHK102), fed overnight with 200 µM sodium palmitate, stained with LipidTOX Deep Red and infected (MOI 5) with mCerulean-producing *L. pneumophila* Δ*legG1* (pNP99). Infected cells were recorded for 60 s each at the times indicated.

https://elifesciences.org/articles/85142/figures#video2

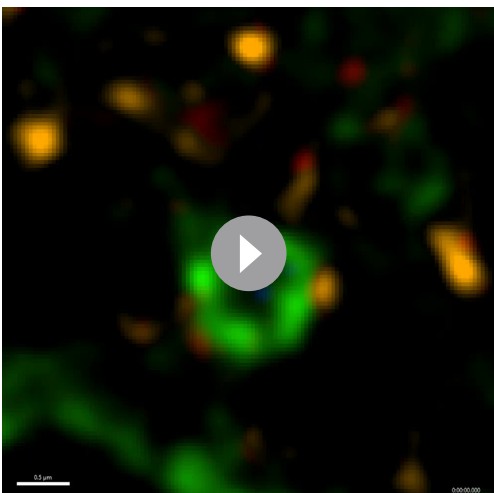

**Video 3.** *D. discoideum* Δ*sey1* infected with *L. pneumophila* JR32. Representative movie of *D. discoideum* Δ*sey1* producing P4C-GFP (pWS034) and mCherry-Plin (pHK102), fed overnight with 200 μM sodium palmitate, stained with LipidTOX Deep Red and infected (MOI 5) with mCerulean-producing *L. pneumophila* JR32 (pNP99). Infected cells were recorded for 60 s each at the times indicated.
https://elifesciences.org/articles/85142/figures#video3

in vitro reconstitution of the LCV-LD interactions using purified LCVs and LDs from either the *D. discoideum* Ax3 parental strain or Δ*sey1* mutant amoeba revealed that the large fusion GTPase Sey1 present in the LD fraction promotes the process in a GTP-dependent manner.

## LDs shed perilipin upon LCV membrane crossing independently of Sey1 and GTP

During our analysis of LCV-LD interactions, we observed that intra-LCV LDs appeared to have lost their perilipin decoration. To analyze in more detail and quantify this observation, we used palmitate-fed *D. discoideum* Ax3 or Δ*sey1* producing P4C-GFP and mCherry-Plin. Upon staining the LDs with LipidTOX Deep Red and infection with mCerulean-producing *L. pneumophila* JR32, we quantified the portion of intra-LCV LDs without perilipin coat (*Figure 6A*). While ca. 90% of the LDs adhering to LCVs from the cytoplasmic side were decorated with perilipin, less than 10% of the LDs in the LCV lumen were decorated with perilipin, regardless of whether the amoeba produced Sey1 or not (*Figure 6B*). Accordingly, most LDs had shed the perilipin coat on their way from the host cell cytoplasm to the lumen of the pathogen vacuole, and this process did not involve the large fusion GTPase Sey1.

To further analyze the shedding process, we mixed LCVs isolated from *D. discoideum* Ax3 producing P4C-GFP with LDs purified from palmitate-fed strain Ax3 producing mCherry-Plin in presence of 5 mM GTP or GDP (*Figure 6C*). Again, ca. 90% of the LDs adhering externally to LCVs were decorated with perilipin, and less than 10% of the LDs in the LCV lumen were decorated with perilipin, regardless of whether GTP or GDP was added (*Figure 6D*). In summary, these results indicate that upon LCV membrane crossing, LDs shed their perilipin coat in a process that does not involve Sey1 or GTP.

## Sey1 and *L. pneumophila* FadL promote intracellular replication and $^{13}C_{16}$-palmitate catabolism

*L. pneumophila* catabolizes palmitate upon growth in broth (*Häuslein et al., 2017*), but it has not been analyzed whether and how the bacteria might use fatty acids during intracellular growth. Based on the observation that LDs fuse with and cross the LCV membrane concomitantly shedding their perilipin coat, LDs might deliver fatty acids to intra-vacuolar *L. pneumophila*. Moreover, fatty acids might be taken up by *L. pneumophila* through the putative outer membrane fatty acid transporter FadL.

**Video 4.** *D. discoideum* Δ*sey1* infected with *L. pneumophila* Δ*legG1*. Representative movie of *D. discoideum* Δ*sey1* producing P4C-GFP (pWS034) and mCherry-Plin (pHK102), fed overnight with 200 μM sodium palmitate, stained with LipidTOX Deep Red and infected (MOI 5) with mCerulean-producing *L. pneumophila* Δ*legG1* (pNP99). Infected cells were recorded for 60 s each at the times indicated.
https://elifesciences.org/articles/85142/figures#video4

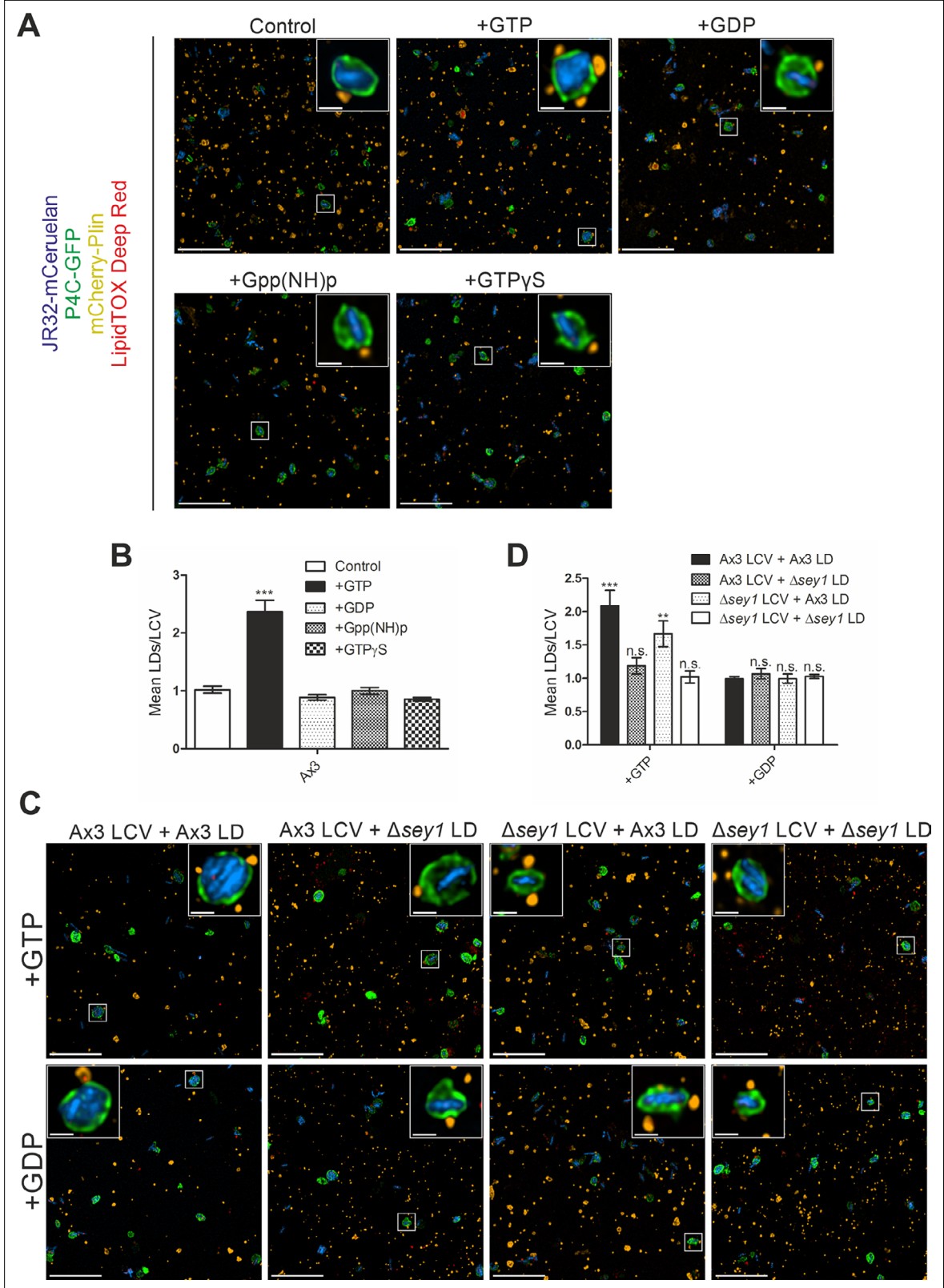

**Figure 5.** Sey1 and GTP promote LCV-LD interactions in vitro. (**A**) Representative fluorescence micrographs of LCVs isolated from *D. discoideum* Ax3 producing P4C-GFP (pWS034), infected (MOI 50, 1 hr) with mCerulean-producing *L. pneumophila* JR32 (pNP99) and mixed with LDs from *D. discoideum* Ax3 producing mCherry-Plin (pHK102) fed overnight with 200 µM sodium palmitate. LCVs and LDs were co-incubated (1 hr, 30 °C) in presence of 5 mM MgCl₂ and 5 mM GTP, GDP, Gpp(NH)p or GTPγS and fixed with PFA prior to imaging. Scale bars: overview (10 µm), inset (1 µm). (**B**) Quantification of

*Figure 5 continued on next page*

*Figure 5 continued*

(**A**), mean number of LDs contacting a single LCV in vitro ($n^{LCVs}$ = 150). Data represent means ± SD of three independent experiments (\*\*\*p<0.001). (**C**) Representative fluorescence micrographs of LCVs isolated from *D. discoideum* Ax3 or Δ*sey1* producing P4C-GFP (pWS034), infected (MOI 50, 1 hr) with mCerulean-producing *L. pneumophila* JR32 (pNP99) and mixed with LDs from *D. discoideum* Ax3 or Δ*sey1* producing mCherry-Plin (pHK102) fed overnight with 200 µM sodium palmitate. LCVs and LDs were co-incubated (1 hr, 30 °C) in presence of 5 mM $MgCl_2$ and 5 mM GTP (**top**) or GDP (**bottom**) and fixed with PFA prior to imaging. Scale bars: overview (10 µm), inset (1 µm). (**D**) Quantification of (**C**), mean number of LDs contacting a single LCV in vitro ($n^{LCVs}$ = 150). Data represent means ± SD of three independent experiments (n.s., not significant; \*\*p<0.01; \*\*\*p<0.001).

The *L. pneumophila* Philadelphia-1 genome comprises one gene that encodes a FadL homolog, *lpg1810*. The amino acid sequence of Lpg1810 is 24% identical to the sequence of *E. coli* FadL, and the highly conserved NPA (asparagine-proline-alanine) motif is also present (amino acids 73–75) (*Figure 7—figure supplement 1*). The *lpg1810* gene and its genomic location are conserved among *L. pneumophila* and *L. longbeachae* strains, and the gene does not seem to be part of an operon. Genes encoding homologs of the acyl-CoA synthetase FadD, and the β-oxidation enzymes FadA, FadB, and FadE are present in these *Legionella* genomes, but no regulatory protein FadR homolog was identified.

To assess the role of *fadL* for the growth of *L. pneumophila*, we constructed a Δ*fadL* deletion mutant strain by double homologous recombination. The Δ*fadL* mutant strain grew like the parental strain JR32 in AYE (ACES yeast extract) broth and MDM (minimal defined medium) (*Figure 7—figure supplement 2*), and up to 200 µM palmitate did not affect the growth of *L. pneumophila* JR32 or Δ*fadL* in AYE broth (*Figure 7—figure supplement 2*).

While the lack of *fadL* did not affect growth of *L. pneumophila* in medium, the Δ*fadL* mutant strain was impaired for intracellular growth in *D. discoideum*, and the growth defect was complemented by inserting the *fadL* gene back into the *L. pneumophila* genome (*Figure 7A*). In agreement with a role of *fadL* for intracellular growth of *L. pneumophila*, a transcriptional $P_{fadL}$-*gfp* reporter construct was expressed in *D. discoideum* throughout infection (*Figure 7—figure supplement 2*).

Next, we tested the effects of overnight palmitate feeding of *D. discoideum* Ax3 or Δ*sey1* on intracellular growth of *L. pneumophila* JR32 or Δ*fadL*. Feeding with palmitate augmented the growth of *L. pneumophila* JR32 in *D. discoideum* Ax3 but did not significantly affect the growth of the Δ*fadL* strain in either *D. discoideum* Ax3 or Δ*sey1* (*Figure 7B*). These results revealed that the host large fusion GTPase Sey1 as well as the putative *L. pneumophila* fatty acid transporter FadL are required for intracellular growth promotion by palmitate and presumably the catabolism of this carbon and energy source.

Finally, to test whether Sey1 and FadL indeed promote the intracellular catabolism of palmitate, we used the isotopologue profiling method. With this approach, the intracellular catabolism by *L. pneumophila* of $^{13}$C-labeled carbon sources via $^{13}C_2$-acetyl-CoA and incorporation of label into the storage compound poly-3-hydroxybutyrate (PHB) is followed by MS (*Eylert et al., 2010*; *Schunder et al., 2014*; *Häuslein et al., 2016*). To test intracellular catabolism of palmitate, calnexin-GFP-producing *D. discoideum* Ax3 or Δ*sey1* were infected with mCerulean-producing *L. pneumophila* JR32 or Δ*fadL*, treated with 200 µM [U-$^{13}C_{16}$]palmitate for 10 hr, lysed and fractionated. For the isotopologue profiling experiments, we generated 3 fractions: fraction 1 (pellet of low-speed centrifugation of infected amoeba = host cell debris), fraction 2 (pellet of high-speed centrifugation = bacteria) and fraction 3 (precipitated supernatant of high-speed centrifugation = cytoplasmic proteins). While the incorporation of label into amino acids was comparable for fraction 1–3, the incorporation of label into 3-hydroxybutyrate (3-HB) was only seen in fraction 2, since 3-HB is specific for bacterial anabolism and not produced by eukaryotic cells. $^{13}$C-Excess (mol%) in key metabolites was analyzed by gas chromatography/MS and is shown for the *L. pneumophila* fraction (*Figure 7C*). This approach revealed a minor enrichment of $^{13}$C label in the amino acids Ala, Asp and Glu; in Glu, the enrichment was less efficient in absence of Sey1. A major enrichment of $^{13}$C-label occurred in the PHB hydrolysis product 3-HB. Interestingly, the enrichment was significantly lower in absence of Sey1 or FadL, and lowest in absence of both, Sey1 and FadL. In summary, these studies revealed that the host large fusion GTPase Sey1 as well as the bacterial outer membrane fatty acid transporter FadL are implicated in intracellular palmitate catabolism by *L. pneumophila*.

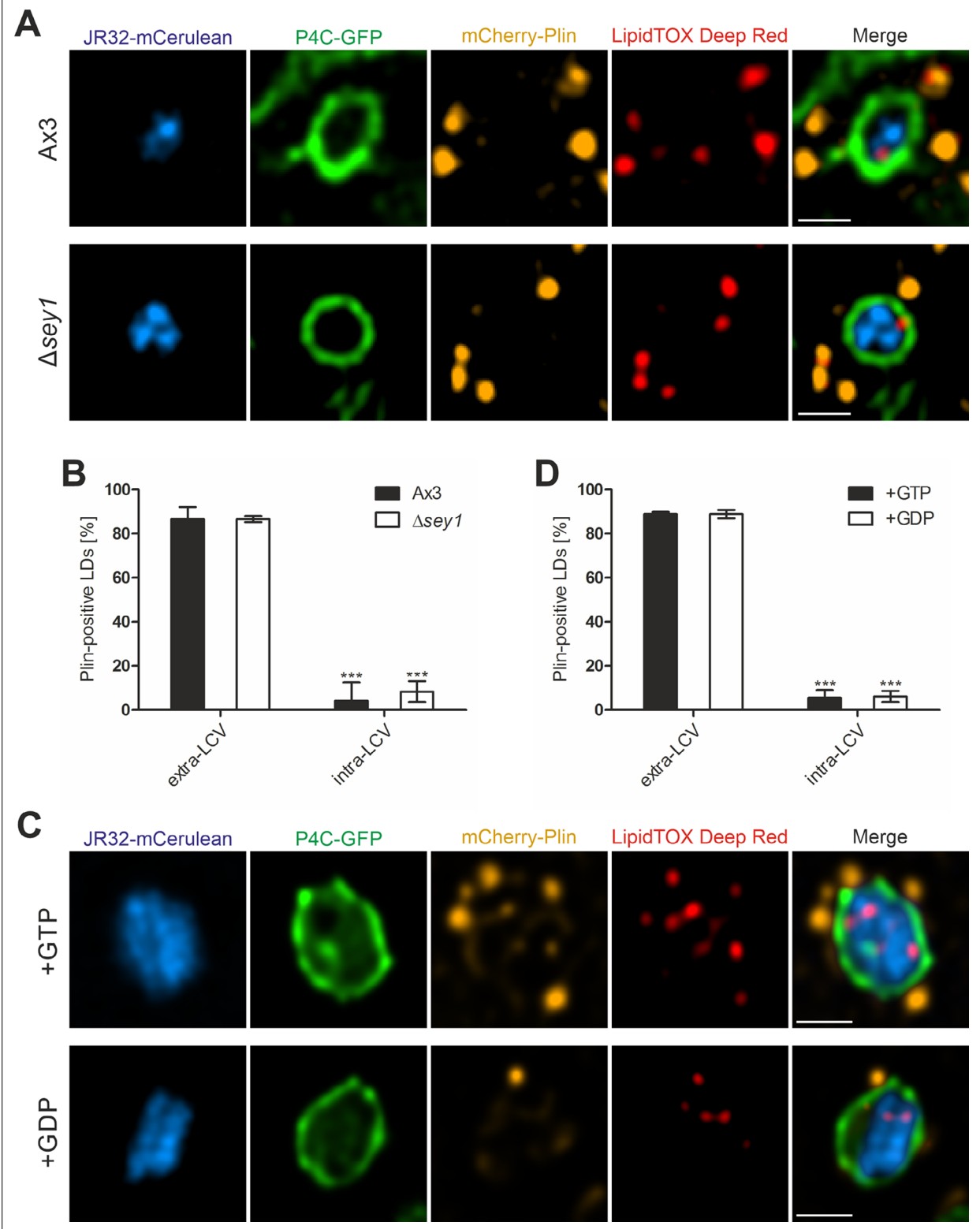

**Figure 6.** LDs shed perilipin upon LCV membrane crossing independently of Sey1 and GTP. (**A**) Representative fluorescence micrographs of *D. discoideum* Ax3 or Δ*sey1* producing P4C-GFP (pWS034) and mCherry-Plin (pHK102), fed overnight with 200 μM sodium palmitate, stained with LipidTOX Deep Red and infected (MOI 5, 1 hr) with mCerulean-producing *L. pneumophila* JR32 (pNP99). Scale bars: 1 μm. (**B**) Quantification of (**A**), percentage of extravacuolar and intravacuolar LDs staining positive for mCherry-Plin ($n^{LDs}$ >24). Data represent means ± SD of three independent experiments (***p<0.001). (**C**) Representative fluorescence micrographs of LCVs isolated from *D. discoideum* Ax3 producing P4C-GFP (pWS034),

*Figure 6 continued on next page*

*Figure 6 continued*

infected (MOI 50, 1 hr) with mCerulean-producing *L. pneumophila* JR32 (pNP99) and mixed with LDs from *D. discoideum* Ax3 producing mCherry-Plin (pHK102) fed overnight with 200 µM sodium palmitate. LCVs and LDs were co-incubated (1 hr, 30 °C) in presence of 5 mM $MgCl_2$ and 5 mM GTP or GDP and fixed with PFA prior to imaging. Scale bars: 1 µm. (**D**) Quantification of (**C**) percentage of extravacuolar and intravacuolar LDs staining positive for mCherry-Plin ($n^{LDs}$ = 180). Data represent means ± SD of three independent experiments (***$p<0.001$).

## Discussion

In this study, we assessed the role of LDs, Sey1 and FadL for intracellular growth of *L. pneumophila*. We reveal that LCVs intimately interact with palmitate-induced LDs in *D. discoideum* (*Figure 1*). The process depends on the large fusion GTPase Sey1 (*Figure 2*), as well as on the *L. pneumophila* Icm/Dot T4SS (*Figure 3*), the effector protein LegG1 (*Figure 4*) and the fatty acid transporter FadL (*Figure 7*). Using dually fluorescence-labeled amoeba, we assessed LCV-LD interactions in vivo by live-cell and fixed-sample microscopy (*Figures 2–4*), or in vitro using reconstituted purified LCVs and LDs (*Figure 5*). The palmitate-induced LDs translocated into the LCV lumen (*Figure 6*), and palmitate was catabolized in an Sey1 and FadL-dependent manner (*Figure 7*). Taken together, our results indicate that Sey1-dependent recruitment of LDs, LCV-LD interactions, LD transfer to the LCV lumen and FadL-dependent catabolism of fatty acids promote intracellular growth of *L. pneumophila* (*Figure 8*).

The LCV-LD interactions are controlled by the *L. pneumophila* Icm/Dot T4SS (*Figure 3*), and the Icm/Dot substrate LegG1 (*Figure 4*). LegG1 belongs to a family of RCC1 repeat domain-containing *L. pneumophila* effectors, which activate the small GTPase RanA at different sites in the cell (LCV, plasma membrane) and consequently stabilize non-centrosomal microtubules (*Rothmeier et al., 2013*; *Swart et al., 2020a*; *Swart et al., 2020c*). Intriguingly, RanA as well as the Ran-binding protein RanBP1 was also identified on LDs (*Supplementary file 1*, *Figure 4—figure supplement 1*). Since RanBP1 only binds to activated, GTP-bound RanA, the small GTPase is likely activated on LDs. Accordingly, LegG1 might not only activate RanA on LCVs (*Rothmeier et al., 2013*; *Swart et al., 2020a*; *Swart et al., 2020c*) but also on LDs, leading to a further stabilization of microtubules, along which the LCV-LD interactions occur. In agreement with this notion, GFP-LegG1 shows a punctate localization in *D. discoideum* and appears to accumulate in the vicinity of LDs (*Figure 4—figure supplement 2*).

Among the more than 300 different *L. pneumophila* effectors, LegG1 and others might (directly) target LDs. An attractive candidate is RalF, which acts as an Arf1 guanine nucleotide exchange factor (GEF), thus activating the small GTPase Arf1 on LCVs (*Nagai et al., 2002*). Activated, GTP-bound Arf1 regulates through perilipin-2 the phospholipid monolayer that stabilizes LDs and controls the cellular availability of neutral lipids for metabolic purposes (*Paulion et al., 2016*). Moreover, upon activation, Arf1 and the coat complex COPI form LD-ER membrane bridges for targeting catabolic enzymes (*Wilfling et al., 2014*). Hence, by acting as an Arf1 GEF, the *L. pneumophila* effector RalF might regulate LD-dependent processes during *L. pneumophila* infection.

While LDs undergo homotypic interactions and fusions, they do not seem to fuse with other organelles. LDs are formed within and released from the ER bilayer. The release is reversible, and accordingly, cytosolic LDs interact with and re-integrate into the ER (*Thiam et al., 2013*; *Wilfling et al., 2013*; *Wilfling et al., 2014*). The process is dependent on the small GTPase Arf1 and the coatomer complex-I (COPI), but otherwise is ill defined, and specific 'integration factors' have not been identified. Using an analogous mechanism, LDs might interact with and integrate into the LCV limiting membrane.

In addition to *L. pneumophila*, several intracellular bacterial pathogens accumulate LDs on their pathogen vacuoles to get access to and metabolize fatty acids (*Walpole et al., 2018*; *Bosch et al., 2021*; *Brink et al., 2021*). Pathogenic *Mycobacterium* species, such as *M. tuberculosis* and *M. leprae*, trigger LD formation during infection, to exploit lipids derived from these organelles as carbon and energy source. *M. tuberculosis* actively evokes LDs in macrophages, thus promoting the 'foamy' appearance of these host cells, to employ lipids as primary energy source in the 'dormant' state during chronic infection (*Singh et al., 2012*; *Mehrotra et al., 2014*). Moreover, LDs accumulate around *Mycobacterium*-containing vacuoles (MCVs) harboring *M. marinum* in *D. discoideum* (*Barisch et al., 2015*; *Barisch and Soldati, 2017a*; *Barisch and Soldati, 2017b*). These LDs cross to the lumen of the MCV and form 'intracytosolic lipid inclusions' (ILIs) within the bacteria. However, during the *D. discoideum*-*M. marinum* encounter, ILI formation and lipid catabolism apparently do not lead to dormancy of the bacteria.

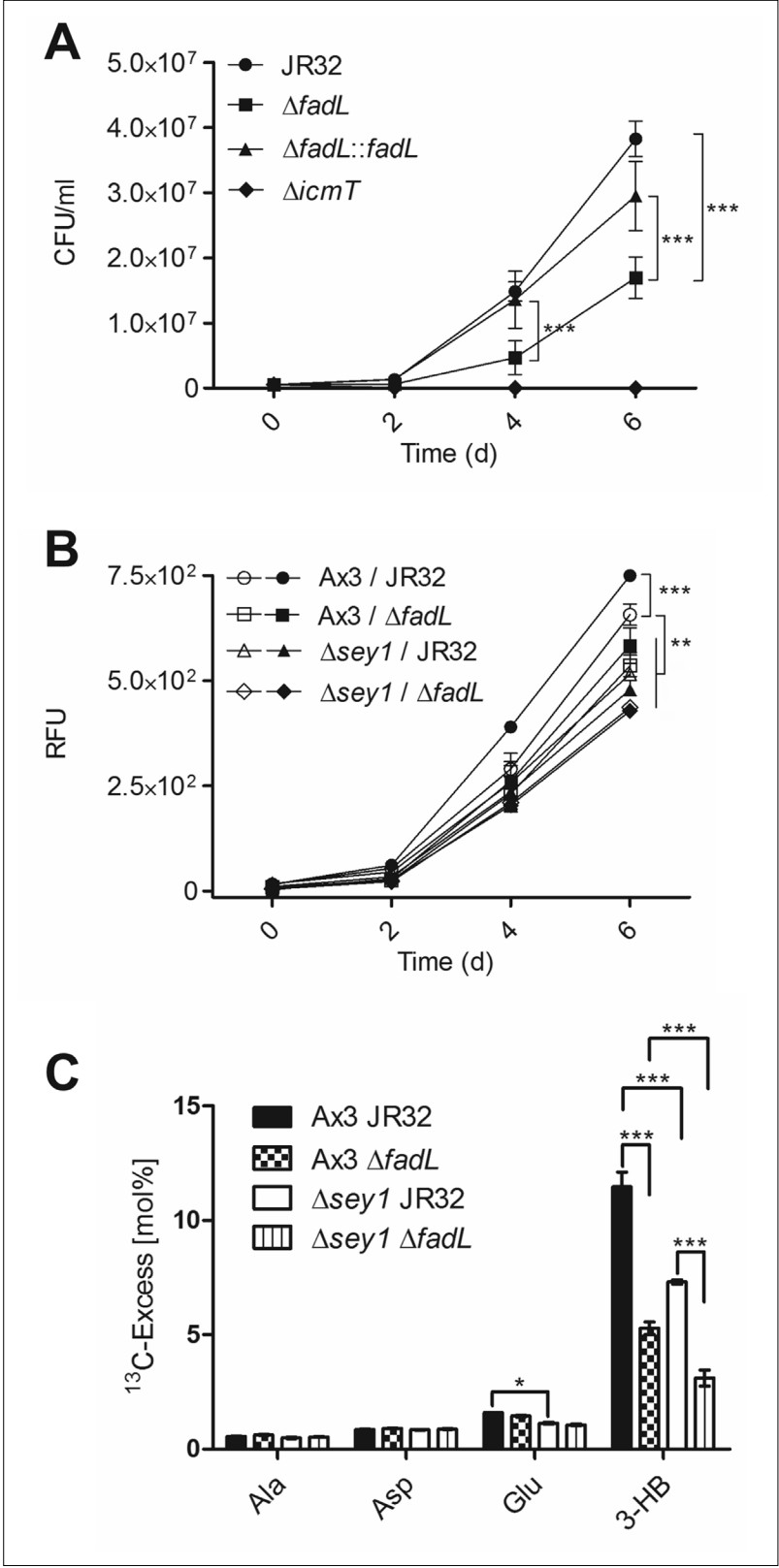

**Figure 7.** Sey1 and *L. pneumophila* FadL promote intracellular replication and $^{13}C_{16}$-palmitate catabolism. (**A**) *D. discoideum* Ax3 amoeba were infected (MOI 1) with *L. pneumophila* JR32, Δ*icmT*, Δ*fadL* or Δ*fadL::fadL* (chromosomal integration of *fadL* in Δ*fadL*), and intracellular bacterial replication was assessed by colony forming units (CFU) for 6 days. Data represent means ± SD of three independent experiments in technical triplicates

*Figure 7 continued on next page*

*Figure 7 continued*

(***p<0.001). (**B**) *D. discoideum* Ax3 or Δ*sey1* were left untreated (**empty symbols**) or fed overnight with 200 μM
sodium palmitate (**filled symbols**) and infected (MOI 1) with GFP-producing *L. pneumophila* JR32 or Δ*fadL* (pNT28)
for 6 days. Intracellular bacterial replication was assessed by quantification of relative fluorescence units (RFU).
Data represent means ± SD of three independent experiments in technical hextuplicates (**p<0.01; ***p<0.001).
(**C**) *D. discoideum* Ax3 or Δ*sey1* producing calnexin-GFP (CnxA-GFP, pAW016) were infected (MOI 50, 1 hr) with
mCerulean-producing *L. pneumophila* JR32 or Δ*fadL* (pNP99) and washed to remove extracellular bacteria. At
5 hr p.i., 200 μM [U-$^{13}$C$_{16}$]palmitate was added to the infected amoeba for 10 hr. The infected cells were lysed and
centrifuged to separate bacteria from cell debris. $^{13}$C-excess (mol%) in key metabolites of the *L. pneumophila*
fraction ('fraction 2') was analyzed by GC/tandem MS. Data show means ± SD of technical triplicates (*p<0.05;
***p<0.001) and are representative for two independent experiments.

The online version of this article includes the following figure supplement(s) for figure 7:

**Figure supplement 1.** Alignment of FadL homologues.

**Figure supplement 2.** Growth of *L. pneumophila* Δ*fadL* in medium and expression of *fadL* in *D. discoideum*.

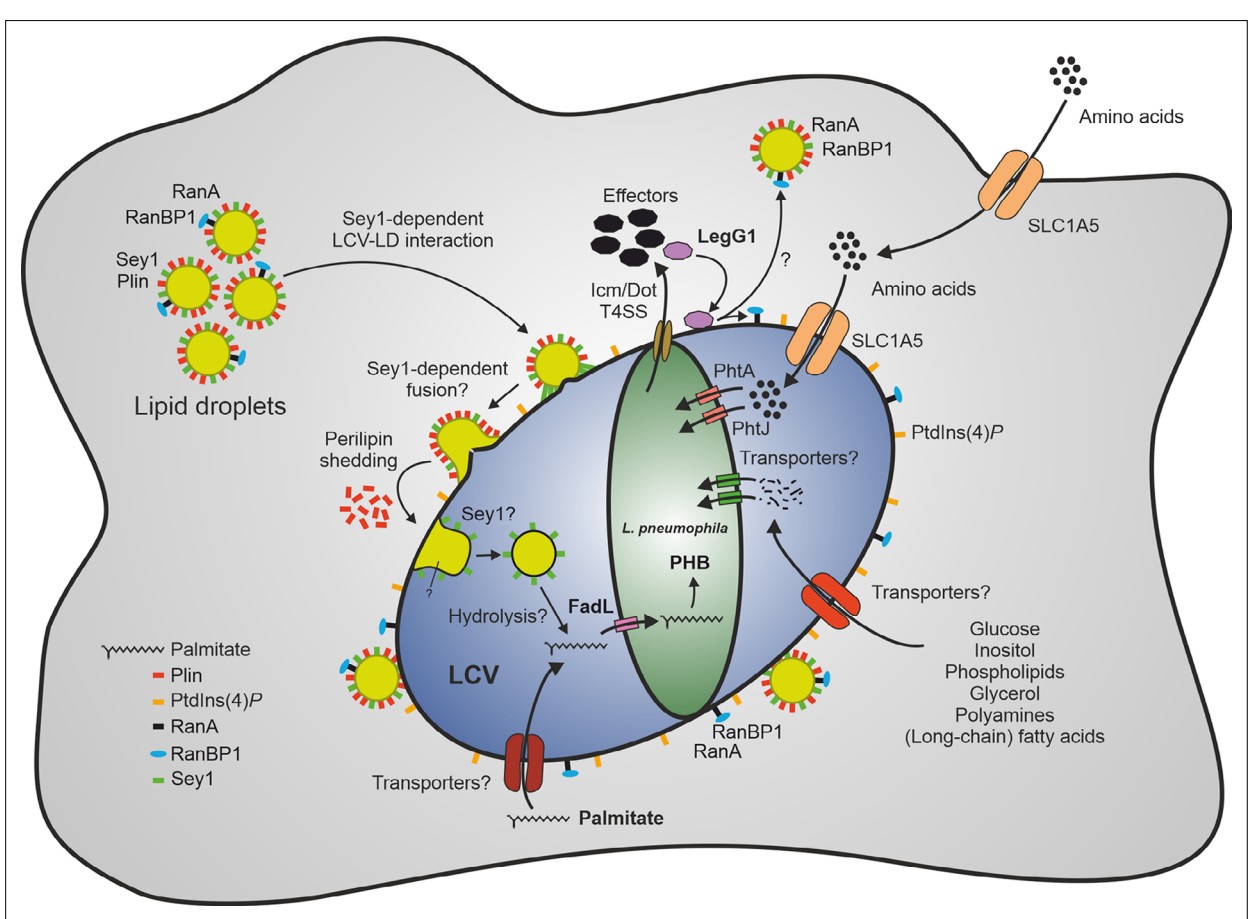

**Figure 8.** Sey1-, LegG1- and FadL-dependent catabolism of palmitate by *L. pneumophila* through lipid droplets. The large fusion GTPase Sey1 and
GTP as well as the *L. pneumophila* RCC1 repeat effector LegG1 promote the recruitment of host LDs to LCVs. LegG1 activates the small GTPase
RanA, leading to accumulation of RanBP1 and microtubule stabilization. Upon intimate contact of LDs with LCVs, perilipin is shed, and LDs cross
the membrane to reach the LCV lumen, where LD constituents (e.g. triacylglycerols) are likely hydrolyzed, and free fatty acids are taken up by *L.
pneumophila*. Palmitate is transported by FadL inside the bacteria and metabolized to acetyl-CoA, which is further aerobically catabolized in the
tricarboxylic acid cycle to CO$_2$ and/or anabolized to the storage compound polyhydroxybutyrate (PHB). Amino acids are transported by the host
transporter SLC1A5 and the *L. pneumophila* transporters PhtA/PhtJ. Additional trans-membrane transporters for lipids (long-chain fatty acids,
phospholipids), sugars (glucose, inositol), alcohols (glycerol), polyamines, and amino acids are likely present in the plasma membrane, LCV membrane
and *L. pneumophila* membrane. LD, lipid droplet; LCV, *Legionella*-containing vacuole; PHB, polyhydroxybutyrate; Plin, perilipin; RanBP1, Ran-binding
protein 1.

The obligate intracellular pathogen *Chlamydia trachomatis* forms a pathogen vacuole termed the inclusion (*Fields and Hackstadt, 2002*). The inclusion accumulates various host lipids, such as sphingomyelin (*Hackstadt et al., 1996*; *Elwell et al., 2011*) and cholesterol (*Carabeo et al., 2003*), and sphingomyelin promotes intracellular bacterial growth (*van Ooij et al., 2000*; *Elwell and Engel, 2012*). The *C. trachomatis* inclusions also capture LDs (*Kumar et al., 2006*), which transverse across the inclusion membrane to the lumen of the pathogen compartment (*Cocchiaro et al., 2008*). Interestingly, LD uptake into *Chlamydia* inclusions proceeds through the formation of an effector protein-mediated LD-inclusion contact site, followed by what appears to be an endocytic uptake leading to membrane-coated, intra-luminal LDs (*Cocchiaro et al., 2008*). LDs seem to be a source of lipids for *C. trachomatis*, but they are not essential as a delivery carrier of fatty acids (*Sharma et al., 2018*). Fatty acids are activated by host long chain fatty acid Acyl-CoA synthases, which are recruited to the inclusion (*Recuero-Checa et al., 2016*). Membrane-coated, intra-luminal LDs have not been observed in the lumen of LCVs, and hence, it is presently unclear how the LDs translocate into the LCV lumen and how the neutral lipids are made available to intra-vacuolar *L. pneumophila*.

*L. pneumophila* FadL promotes the catabolism of palmitate (*Figure 7*), but overall, the degradation of lipids by *Legionella* is not well understood. Analogously to *M. marinum* (*Barisch and Soldati, 2017b*), *L. pneumophila* might hydrolyse triacylglycerols; however, rather than building up ILIs, *L. pneumophila* accumulates the storage compound PHB (*Figure 7*). *L. pneumophila* produces at least 19 phospholipases (15 phospholipases A, three phospholipases C, and one phospholipase D), which act as virulence factors during intracellular replication (*Kuhle and Flieger, 2013*), and some of which might metabolize lipids stored in LDs. While some of the lipases are T2SS substrates and thus are presumably secreted to the LCV lumen, others are Icm/Dot T4SS substrates and translocated to the host cytoplasm. Type II-secreted phospholipases comprise phospholipases A, such as the GDSL lipase family members PlaA and PlaC, or PlaB and PlaD (*Bender et al., 2009*; *Lang et al., 2017*), as well as the zinc metallophospholipases C, such as PlcA and PlcB (*Aurass et al., 2013*). PlcC (CegC1/Lpg0012) is a type IV (Icm/Dot)-translocated zinc metallophospholipases C (*Aurass et al., 2013*). Alternatively, or additionally, some of these phospholipases might be involved in degrading the limiting LCV membrane to promote the escape of *L. pneumophila* from the pathogen vacuole and the host cell. In agreement with this notion, *L. pneumophila* triple mutant strains lacking phospholipases A or C were impaired for LCV rupture in *D. discoideum*, indicating that phospholipases A and C are indeed involved in membrane lysis (*Striednig et al., 2021*). Further studies will address the role of these phospholipases for *L. pneumophila* LD utilization and lipid metabolism. In any case, lipid metabolism and LDs seem to be important for optimal intracellular replication of *L. pneumophila*, but not essential for bacterial replication and infection.

Mechanistically, the fusion of LDs with distinct pathogen vacuoles is poorly understood (*Figure 1B*). In particular, it is unknown whether and how LDs tether and cluster on LCVs and how they cross the pathogen vacuole membrane and translocate inside the pathogen vacuole. As outlined above, RanA and RanBP1 were identified with high probability on LDs isolated from the *D. discoideum* parental strain Ax3 as well as from Δ*sey1* mutant amoeba (*Supplementary file 1*), and the *L. pneumophila* RanA activator LegG1 promotes LCV-LD interactions (*Figure 4*). The underlying mechanism of other factors implicated in LCV-LD interactions is less clear. Given the role of Sey1 for the recruitment, accumulation and contact time of LDs on LCVs, the large fusion GTPase might also play a role in later steps of LCV-LD fusion and membrane crossing. Overall, the proteomics analysis revealed similar proteomes of the LDs isolated from the *D. discoideum* parental strain Ax3 versus Δ*sey1* mutant amoeba (*Supplementary file 1*). Sey1 was identified with low probability by proteomics on LDs purified from the *D. discoideum* Ax3 parental strain, and Sey1 does not seem to significantly affect the number and size of LDs in *D. discoideum* (*Figure 3—figure supplement 1*). Accordingly, the fatty acid and lipid content of LDs isolated from these two different *D. discoideum* strains might likely also be similar. Taken together, these findings are in agreement with a direct role of Sey1 on LDs for the LCV-LD fusion process. However, Sey1 might also indirectly affect LCV-LD fusion. *D. discoideum* Δ*sey1* mutant amoeba show pleiotropic phenotypes, such as an altered ER architecture, a partially disrupted tubular ER network and defects in intracellular proteolysis, cell motility, and growth on bacterial lawns as well as an impaired LCV expansion and intracellular replication of *L. pneumophila* (*Hüsler et al., 2021*). On the other hand, the loss of *sey1* does not have major consequences for organelles other than the ER or the architecture of the secretory and endocytic pathways. Since Sey1

affects a number of cellular functions, the large GTPase might also have an effect on the formation and composition of LDs.

The LD protein perilipin was shed upon uptake of LDs into LCVs (*Figure 6*). Perilipin reversibly interacts with the LD surface via amphipathic helices (*Kory et al., 2016*), and therefore, it might be rather easily shed and distributed to the cytoplasm upon LD translocation to the LCV lumen. The mechanism and biological function of perilipin shedding upon pathogen vacuole transfer of the LDs is unknown. Future investigations using the LCV-LD model system for in vivo and in vitro studies by confocal microscopy and reconstitution approaches will provide further insights on the underlying cell biological and infection biological processes.

## Materials and methods

**Key resources table**

| Reagent type (species) or resource | Designation | Source or reference | Identifiers | Additional information |
|---|---|---|---|---|
| Gene (*Legionella pneumophila* Philadelphia-1) | *lpg1810/fadL* | GenBank | AE017354.1 | |
| Strain, strain background (*Legionella pneumophila*) | ER01 (Δ*legG1*) | *Rothmeier et al., 2013* | | JR32 *legG1*::Kan$^R$ |
| Strain, strain background (*Legionella pneumophila*) | GS3011 (Δ*icmT*) | *Segal and Shuman, 1998* | | JR32 *icmT3011*::Kan$^R$ |
| Strain, strain background (*Legionella pneumophila*) | JR32 | *Sadosky et al., 1993* | | Derivative of wild-type *Legionella pneumophila* strain Philadelphia-1 |
| Strain, strain background (*Legionella pneumophila*) | PS01 (Δ*fadL*) | This study | | JR32 *fadL*::Kan$^R$ |
| Strain, strain background (*Escherichia coli*) | TOP10 | Invitrogen, Thermo Fisher Scientific | | |
| Strain, strain background (*Dictyostelium discoideum*) | Ax3 | *Loovers et al., 2007* | | Parental strain |
| Strain, strain background (*Dictyostelium discoideum*) | Δ*sey1* | *Hüsler et al., 2021* | | Ax3, insertion in gene DDB_G0279823, Bls$^R$ |
| Antibody | anti-SidC (rabbit polyclonal) | *Weber et al., 2006* | | 1:3000 |
| Antibody | anti-rabbit IgG MACS micro-beads (goat polyclonal) | Miltenyi Biotec | Cat# 130-048-602 | 20 µl magnetic bead slurry per 0.5 ml concentrated cell homogenate |
| Chemical compound, drug | LipidTOX Deep Red | Invitrogen, Thermo Fisher Scientific | Cat# H34477 | 1:200 – 1:1000 |

### Bacteria, cells, and reagents

Bacterial strains and *D. discoideum* strains used are listed in *Supplementary file 2*. *L. pneumophila* strains were grown for 3 days on charcoal yeast extract (CYE) agar plates, buffered with *N*-(2-acetamido)–2-aminoethane sulfonic acid (ACES) at 37 °C. Liquid cultures in ACES yeast extract (AYE) medium were inoculated at an $OD_{600}$ of 0.1 and grown at 37 °C for 21 hr to an early stationary phase ($2 \times 10^9$ bacteria/ml). Chloramphenicol (Cam; 5 µg/ml) was added when required.

Growth of *L. pneumophila* strains was assessed in 3 ml AYE or 3 ml minimal defined medium (MDM) at an initial $OD_{600}$ of 0.1 and incubated on a rotating wheel (80 rpm, 37 °C) for 18 hr or 28 hr. The cultures were then diluted in the respective medium to an $OD_{600}$ of 0.2 in a 96-well plate (200 µl/well) and incubated at 37 °C while orbitally shaking. The $OD_{600}$ was measured in triplicates using a micro-titer plate reader to assess growth in AYE (Synergy H1 Hybrid Reader, BioTek) or MDM (Cytation 5 Cell Imaging Multi-Mode Reader, BioTek).

For GFP reporter assays, bacterial overnight cultures were diluted to an initial $OD_{600}$ of 0.2. In a black 96-well plate, 200 µl diluted over-night culture per well was incubated at 37 °C while orbitally shaking. Bacterial growth and GFP production were monitored in triplicates by measuring the $OD_{600}$ and fluorescence (excitation, 485 nm; emission, 528 nm; gain, 50) using a microtiter plate reader (Cytation 5 Cell Imaging Multi-Mode Reader, BioTek). Values are expressed as relative fluorescence units (RFU) or $OD_{600}$.

*D. discoideum* strains were grown at 23 °C in HL-5 medium (ForMedium). Cells were maintained every 2–3 days by rinsing with fresh HL-5 and by transferring 1–2% of the volume to a new T75 flask containing 10 ml medium. Cells were strictly maintained below 90% confluence. Transformation of axenic *D. discoideum* amoeba was performed as previously described (*Weber and Hilbi, 2014a*; *Weber et al., 2014b*). Geneticin (G418, 20 µg/ml) and hygromycin (50 µg/ml) were added when required.

LDs were stained with LipidTOX Deep Red (Invitrogen, Thermo Fisher Scientific), a neutral dye that has a high affinity for neutral LDs.

## Molecular cloning

The plasmids used and generated in this study are listed in *Supplementary file 2*. Cloning was performed according to standard protocols and plasmids were isolated using the NucleoSpin Plasmid kit (Macherey-Nagel). DNA fragments were amplified using Phusion High Fidelity DNA polymerase (NEB) and the oligonucleotides listed in *Supplementary file 3*. FastDigest restriction enzymes (Thermo-Fisher) were used for plasmid digestion and Gibson assembly was performed using the NEBuilder HiFi DNA assembly kit (NEB). All constructs were verified by DNA sequencing.

To construct plasmid pLS187, the gene region of *ranBP1* ($P_{ranBP1}$) was amplified from pSU26 (*Rothmeier et al., 2013*) using the primer pair oLS296/oLS297. The PCR product was purified as described above and cloned into pDM323 (*Veltman et al., 2009*), previously digested with *Bgl*II and *Spe*I. To construct pLS221 and pLS222, the gene region of *ranA* ($P_{ranA}$) or *ranBP1* ($P_{ranBP1}$) was amplified from pSU17 or pLS187, using the primer pairs oLS333/oLS334 or oLS296/oLS297. The PCR products were purified as described above and cloned into the plasmids pDM1044 (*Barisch et al., 2015*) or pDM329 (*Veltman et al., 2009*), digested with *Bgl*II and *Spe*I. To construct plasmid pLS117 (pDM317-*gfp-legG1*), the *legG1* gene was amplified from pER017 (*Swart et al., 2020c*) using the primers oLS178/oLS180, cut with *Bgl*II and *Spe*I and ligated into the same sites of pDM317.

To construct the GFP reporter plasmid pPS003, the promoter region of *fadL* ($P_{fadL}$) was amplified from genomic DNA of *L. pneumophila* using the primers oPS013 and oPS015. The PCR product was purified using the NucleoSpin Gel and PCR Clean-up kit (Macherey-Nagel) and cloned into the plasmid pCM009 (*Schell et al., 2016*), which was previously digested with *Sac*I and *Xba*I to remove $P_{flaA}$. Thus, $P_{flaA}$-*gfp* was replaced by $P_{fadL}$-*gfp* in the final construct.

## Construction of the *L. pneumophila* Δ*fadL* mutant

The Δ*fadL* mutant was generated by double homologous recombination as described (*Tiaden et al., 2007*), replacing the *fadL* gene by a kanamycin (Kan) resistance cassette. The primers oPS001 and oPS002 were used to amplify from *L. pneumophila* genomic DNA the upstream flanking region of *fadL*, oPS005 and oPS006 to amplify the downstream flanking region of *fadL* (939 bp each). The Kan resistance cassette was amplified using the primers oPS003 and oPS004, and the plasmid pUC4K (Amersham) as a template. The three amplified fragments were cloned into *Bam*HI-digested pUC19 in a 3-way ligation and amplified in one piece using the primers oPS010 and oPS011. To yield the allelic exchange vector pPS002, the amplified sequence was cloned into the *Bam*HI-digested suicide vector pLAW344 (*Wiater et al., 1994*), which allows counter-selection with the *sacB* gene. *L. pneumophila* JR32 was transformed with pPS002 by electroporation and grown on CYE plates supplemented with 5 µg/ml Cam. Individual clones were picked, grown in AYE medium overnight and plated on CYE plates supplemented with 20 µg/ml Kan and 20 mg/ml sucrose, or 5 µg/ml Cam. After incubation at 37 °C for 2 days, clones growing on the CYE/Kan/sucrose plate were picked and purified by dilution streaking. This selection process was repeated until growth on CYE/Cam plates was no longer observed. Double-cross-over events and thus correct insertion of the Kan resistance cassette in the genome of the deletion mutant were confirmed by PCR and sequencing.

For complementation of the intracellular replication phenotype of Δ*fadL*, the *fadL* gene was re-introduced into the genome of the Δ*fadL* mutant strain by co-integration of the suicide plasmid pPS013. This plasmid was constructed by amplification of *fadL* together with its 3' and 5' flanking regions from *L. pneumophila* genomic DNA using the primers oPS011 and oPS045 and cloning into *Bam*HI-digested pPS002. Transformants were plated on CYE plates supplemented with 5 µg/ml Cam to select for uptake and integration of pPS013.

## Visualization and quantification of *fadL* expression

To assess *fadL* expression in intracellular *L. pneumophila*, *D. discoideum* were infected with *L. pneumophila* JR32 harboring pPS003. Amoeba were counted in a Neubauer improved counting chamber, 0.1 mm depth (Marienfeld-Superior) and $1\times10^6$ *D. discoideum* per well were seeded one day before infection into 6-well plates in 2 ml HL-5. *L. pneumophila* overnight cultures were prepared as described above and grown on a rotating wheel at 37 °C for 21–22 hr to early stationary phase ($OD_{600}$ ca. 5.0, ~$2 \times 10^9$ bacteria/ml). The day of infection, *L. pneumophila* cultures were checked for motility and filamentation, and $OD_{600}$ was measured to calculate the bacterial concentration. *D. discoideum* cells were infected (MOI 5), centrifuged (10 min, 450 *g*, RT), and incubated at 25 °C. After 1.5–2 hr, *D. discoideum* cells were washed four times with HL-5, and incubated further at 25 °C. For the 30 min p.i. time point, amoeba were washed after 30 min and immediately processed further. Infected amoeba were collected into 2 ml Eppendorf tubes 30 min, 6 hr, 24 hr, 32 hr, or 48 hr p.i., and centrifuged (5 min, 2000 *g*, RT) to remove the medium. For microscopy, the cells were fixed in 4% PFA (1 hr, RT), washed with DPBS, and permeabilized with ice-cold methanol for 10 min. Subsequently, cells were stained with 1 µg/ml DAPI (1 hr, RT), washed twice with DPBS, resuspended in 30 µl DPBS, and embedded in 0.5% agarose in an 18-well ibidi dish.

For flow cytometry, cells were lysed in lysis buffer (150 mM NaCl, 0.1% Triton X-100) for 15 min at RT and centrifuged (5 min, 2000 *g*, RT). Supernatant was removed and isolated intracellular bacteria were fixed in 4% PFA (1 hr, RT). The samples were stained with 1 µg/ml DAPI (1 hr, RT), washed twice with DPBS, and resuspended in 500 µl DPBS.

## Intracellular *L. pneumophila* replication

Intracellular replication of *L. pneumophila* JR32, Δ*icmT* and Δ*fadL* in *D. discoideum* amoeba was analyzed by colony-forming units (CFU) as well as by increase in relative fluorescence units (RFU). To determine CFU, *D. discoideum* Ax3 or Δ*sey1* amoeba were seeded at a density of $1\times10^5$ cells/ml in cell culture-treated 96-wells plates (VWR) and cultivated at 23 °C in HL-5 medium. Afterwards, the amoeba were infected (MOI 1) with early stationary phase *L. pneumophila* JR32, Δ*icmT*, Δ*fadL*, or Δ*fadL::fadL* (chromosomal integration of *fadL* in Δ*fadL*) diluted in MB medium (**Solomon and Isberg, 2000**), centrifuged (450 *g*, 10 min, RT) and incubated at 25 °C for the time indicated (96-wells plate was kept moist by addition of dd$H_2$O in surrounding wells). The cells were lysed with 0.8% saponin (Sigma-Aldrich) for 10 min at RT, and dilutions were plated on CYE/Cam agar plates and incubated at 37 °C for 3 days. CFU were assessed every 2 days using an automated colony counter (CounterMat Flash 4000, IUL Instruments, CounterMat software), and the number of CFU (per ml) was calculated.

To determine increase in fluorescence derived from intracellular replication of GFP-producing *L. pneumophila*, *D. discoideum* Ax3 or Δ*sey1* were seeded at a density of $1\times10^5$ cells/ml in cell-culture-treated 96-wells plates (VWR) and cultivated overnight at 23 °C in HL-5 medium supplemented with 200 µM sodium palmitate. A range of different concentrations of sodium palmitate (100–800 µM) were previously tested, and 200 µM sodium palmitate appeared to be the most effective in increasing intracellular replication of *L. pneumophila* (**Figure 1A**). The cells were infected (MOI 1) with early stationary phase GFP-producing *L. pneumophila* JR32 or Δ*fadL* diluted in MB medium, centrifuged (450 *g*, 10 min, RT) and incubated for 1 hr at 25 °C. Afterwards, the infected amoeba were incubated at 25 °C for the time indicated (96-wells plate was kept moist by addition of dd$H_2$O in surrounding wells). Increase in GFP fluorescence was assessed every 2 days using a microtiter plate reader (Synergy H1, Biotek).

## In vitro reconstitution assays

LCVs from *D. discoideum* Ax3 and Δ*sey1* mutant amoeba were purified as previously described (**Urwyler et al., 2010**). Briefly, *D. discoideum* Ax3 or Δ*sey1* producing P4C-GFP (pWS034) were seeded in three T75 flasks per sample one day prior to experiment to reach 80% confluency. The amoeba were infected (MOI 50, 1 hr) with *L. pneumophila* JR32 producing mCerulean (pNP99) grown to stationary phase (21 hr liquid culture). Subsequently, the cells were washed with SorC buffer (2 mM $Na_2HPO_4$, 15 mM $KH_2PO_4$, 50 µM $CaCl_2\times2$ $H_2O$, pH 6.0) and scraped in homogenization buffer (20 mM HEPES, 250 mM sucrose, 0.5 mM EGTA, pH 7.2) containing a protease inhibitor cocktail tablet (Roche) (**Derré and Isberg, 2004**). Cells were homogenized using a ball homogenizer (Isobiotec) with an exclusion size of 8 µm and incubated with an anti-SidC antibody followed by a secondary goat anti-rabbit

antibody coupled to magnetic beads. The LCVs were separated using magnetic columns and further purified by density gradient centrifugation as described (*Hoffmann et al., 2013*).

LDs from *D. discoideum* Ax3 and Δ*sey1* mutant amoeba were purified using an update of a previously published protocol (*Du et al., 2013*). *D. discoideum* Ax3 or Δ*sey1* producing mCherry-Plin (pHK102) were seeded in three T150 flasks per sample one day prior to experiment to reach 80% confluency and stimulated overnight (16 hr) with 200 µM sodium palmitate to induce LDs formation. The next day, the cells were recovered by scraping, washed twice with SorC buffer and once in STKM buffer (50 mM Tris, pH 7.6, 25 mM KCl, 5 mM MgCl₂, 0.25 M sucrose) supplemented with a protease inhibitor cocktail tablet (Roche). Successively, cells were resuspended in 1 ml STKM buffer (supplemented with protease inhibitor cocktail tablet) and homogenized by 9–10 passages through a ball homogenizer (Isobiotec) with an exclusion size of 8 µm. Cell homogenate was then centrifuged (1000 $g$, 30 min, 4 °C), and the supernatant containing LDs was separated from the nuclei pellet. The post-nuclear supernatant was adjusted to 0.8 M sucrose and loaded in the middle of a step gradient ranging from 0.1 to 1.8 M sucrose in STKM buffer and centrifuged at 50,000 $g$ for 1 hr at 4 °C in a TLA-120.1 fixed-angle rotor (Beckman Coulter). LDs formed a white cushion on top of the tube and were collected by aspiration of the upper fraction of the gradient (125 µl).

Isolated LDs and LCVs were combined in vitro to reconstitute and biochemically analyze LCV-LD interactions. Isolated LCVs (300 µl from gradient) were diluted with 1 ml homogenization buffer with protease inhibitors, added on top of sterile poly-L-lysine (Sigma) coated coverslips in a 24-well plate and centrifuged at 600 $g$ for 10 min at 4 °C. Supernatant was aspirated, and isolated LDs (100 µl from gradient), together with 100 µl STKM with protease inhibitors, 5 mM MgCl₂ and 5 mM GTP/GDP/ Gpp(NH)p/GTPγS (Sigma) were appropriately mixed and added on top of the LCV-coated coverslips. The reconstitution plate was then centrifuged (600 $g$, 10 min, 4 °C) and incubated for 1 hr at 30 °C. Following incubation and supernatant aspiration, the samples were fixed with 2% PFA for 1 hr at RT, washed twice with SorC, stained with LipidTOX Deep Red (1:200 in SorC; Invitrogen, Thermo Fisher Scientific) for 30 min at RT in the dark, and finally mounted on microscopy slides.

## Confocal microscopy of bacteria, infected cells, and LCV-LD interactions in vitro

For infection assays for confocal microscopy, *D. discoideum* strains producing the desired fluorescent probes were harvested from approximately 80% confluent cultures, seeded at 1×10⁵ cells/ml in 6-well plates (Corning) or 8-well µ-slides (for live-cell experiments) (ibidi) and cultivated overnight at 23 °C in HL-5 medium supplemented with 200 µM sodium palmitate. Infections (MOI 5) were performed with early stationary phase cultures of *L. pneumophila* JR32, Δ*icmT* or Δ*legG1* mutant strains harboring pNP99 (mCerulean), diluted in HL-5, and synchronized by centrifugation (450 $g$, 10 min, RT) (*Rothmeier et al., 2013*). Subsequently, infected cells were washed three times with HL-5 and incubated at 25 °C for the time indicated. Finally, infected amoeba were recovered from the six-well plates, fixed with 4% PFA for 30 min at RT, stained with LipidTOX Deep Red (1:1000 in SorC; Thermo-Fisher) for 30 min in the dark, transferred to eight-well µ-slides and embedded under a layer of PBS/0.5% agarose before imaging. For live-cell experiments, amoeba were stained prior to infection with LipidTOX Deep Red (1:200 in LoFlo medium) for 30 min in the dark and were directly imaged in LoFlo (ForMedium) in the eight-well µ-slides after washing.

Confocal microscopy of bacteria grown in liquid culture, fixed or live infected cells, and LCV-LD interactions in vitro was performed as described (*Finsel et al., 2013*; *Rothmeier et al., 2013*; *Steiner et al., 2017*; *Weber et al., 2018*) using a Leica TCS SP8 X confocal laser scanning microscope with the following setup: white light laser, 442 nm diode and HyD hybrid detectors for each channel. Pictures were taken using a HC PL APO CS2 63×/1.4 oil objective with Leica Type F immersion oil and analyzed with Leica LAS X software. Settings for fluorescence imaging were as following: DAPI (excitation 350 nm, emission 470 nm), mCerulean (excitation 442 nm, emission 469 nm), eGFP (excitation 488 nm, emission 516 nm), mCherry (excitation 568 nm, emission 610 nm), LipidTOX Deep Red (excitation 637 nm, emission, 655 nm). Images and movies were captured with a pinhole of 1.19 Airy Units (AU) and with a pixel/voxel size at or close to the instrument's Nyquist criterion of approximately 43×43×130 nm (xyz). Scanning speed of 400 Hz, bi-directional scan and line accumulation equal 2 were used to capture still images (*Figure 2D*, *Figure 3*, *Figure 4D*, *Figures 5–6*, *Figure 4—figure supplements 1–2*, *Figure 7—figure supplement 2*). Resonant scanner (scanning speed: 8000 Hz) and

line average equal 4 were used to capture movies (*Figure 2A*, *Figure 4A*, *Figure 2—figure supplement 1*, *Videos 1–4*).

For image processing, all images and movies were deconvolved with Huygens professional version 19.10 (Scientific Volume Imaging, The Netherlands, http://svi.nl) using the CMLE algorithm with 40 iterations and 0.05 quality threshold. Signal to noise ratios were estimated from the photons counted for a given image. Single images, Z-stacks and movies were finalized and exported with Imaris 9.5.0 software (Bitplane, Switzerland) and analyzed using ImageJ software (https://imagej.nih.gov/ij/). An LCV-LD 'contact' was defined as the visual association of the signal on the LCV-limiting membrane (P4C-GFP, AmtA-GFP) with the LD mCherry-Plin signal, and only Plin/LipidTOX Deep Red-positive LDs were considered for LCV-LD contacts.

## Electron microscopy

Infection assays for electron microscopy were performed with *D. discoideum* Ax3 grown directly on grids as previously described (*Medeiros et al., 2018*). Briefly, the amoeba ($5 \times 10^5$ per well) were seeded onto EM gold finder grids (Au NH2 R2/2, Quantifoil) and incubated for 1 hr, to allow the cells to attach to the grids. Cells were infected at an MOI of 100 and were vitrified at different time points (30 min and 3 hr p.i.).

The vitrification of infected host cells was done by plunge-freezing as described previously (*Weiss et al., 2017*; *Medeiros et al., 2018*). In short, gold finder grids (Au NH2 R2/2, Quantifoil) containing infected amoeba were vitrified by back-blotting (Teflon, 2×3–7 s). All grids were plunge-frozen in liquid ethane-propane (37%/63% v/v) using a Vitrobot (Thermo-Fisher) and stored in liquid nitrogen until further use.

Cryo-focused ion beam (cryoFIB) milling was used to prepare samples of plunge-frozen infected amoeba for imaging by cryo-electron tomography (cryoET) (*Marko et al., 2007*). Frozen grids with infected cells were processed as previously described (*Medeiros et al., 2018*) using a Helios Nano-Lab600i dual beam FIB/SEM instrument (Thermo-Fisher). Briefly, lamellae with ~2 µm thickness were generated first (at 30 kV and ~400 pA). The thickness of the lamellae (final ~200 nm) was then gradually reduced using decreasing ion beam currents (final ~25 pA). Lastly, lamellae were examined at low voltage by cryo-electron microscopy (cryoEM) imaging (3 kV,~0.17 nA) to visualize intracellular bacteria. CryoFIB-processed grids were unloaded and stored in liquid nitrogen until further use.

Frozen grids were examined by cryoEM and cryoET (*Weiss et al., 2017*). Data were collected on a Titan Krios TEM (Thermo-Fisher) equipped with a Quantum LS imaging filter and K2 Summit (Gatan). The microscope was operated at 300 kV, and the imaging filter was set to a slit width of 20 eV. The pixel size at the specimen level ranged from 3.45 to 5.42 Å. Tilt series of the lamellae were recorded from –60° to +60° with 2° increments and –8 µm defocus. The total dose of a tilt series was 75 e⁻/Å² (intracellular *L. pneumophila*). Tilt series and 2D projection images were acquired automatically using SerialEM (*Mastronarde, 2005*). Three-dimensional reconstructions and segmentations were generated using the IMOD program suite (*Mastronarde, 2005*).

## Flow cytometry

Intracellular bacteria recovered from infected amoeba or bacterial strains grown in liquid culture were analyzed with an LSRFortessa II flow cytometer (BD Biosciences). Gating was performed as follows: Bacterial cells were identified by forward scatter (FSC) versus side scatter (SSC) gating. The threshold for FSC and SSC was set to 200, and 10,000 events were acquired per sample. The bacteria were further examined for their DAPI (450 nm) and GFP (488 nm) fluorescence. DAPI-stained JR32 and JR32 harboring pNT28 for constitutive GFP production were used as references for gating of the GFP-positive subpopulation in the samples. Thus, the percentage of GFP-positive bacteria was calculated. Data processing was performed using the FlowJo software.

To assess cell viability after sodium palmitate treatment, the Zombie Aqua fixable viability kit (BioLegend) was used. *D. discoideum* cells were grown and treated with sodium palmitate overnight (final concentration 100, 200, 400, and 800 µM). Cells treated for 1 hr with 70% sterile-filtered ethanol (EtOH) served as positive control for cell death. The cells were then harvested in ice-cold SorC and stained for 30 min in the dark with 50 µl Zombie Aqua dye diluted 1:500 in SorC. Cells were washed once with HL-5, centrifuged, washed once with SorC, centrifuged, and fixed with 4% PFA for 30 min at RT. After centrifugation, the cells were resuspended in 500 µl SorC. Subsequently, cells were subjected

to flow cytometry analysis (BD FACS Canto II). Gates were set according to FSC/SSC properties, and 10,000 events were collected for each sample.

## Comparative proteomics of purified LDs

StrataClean beads for protein precipitation were incubated in 37% HCl for 6 hr at 100 °C, centrifuged (3500 $g$, 5 min, RT) and washed twice to remove residual HCl. LDs were harvested from palmitate-fed mCherry-Plin-producing *D. discoideum* Ax3 or Δ*sey1* mutant amoeba, purified by sucrose gradient centrifugation (*Figure 4—figure supplement 1*). To extract LD-bound proteins, LDs purified by sucrose density gradient centrifugation were lysed by freeze/thawing and by sonication (50% power, 3 s pulse for 2 min, 4 °C). Extracted LDs protein sample solutions were then incubated with Strata-Clean beads in an over-head shaker for 2 hr at 4 °C. Following incubation, the beads were centrifuged (10,000 $g$, 45 min, 4 °C), and the resulting pellet was washed once in distilled water (16,229 $g$, 10 min, 4 °C). After supernatant removal, the bead pellet was dried in a vacuum centrifuge for 30 min and stored at 4 °C.

StrataClean-bound LD proteins were resolved by 1D-SDS-PAGE, the gel lanes were excised in ten equidistant pieces and subjected to trypsin digestion (*Bonn et al., 2014*). For the subsequent LC-MS/MS measurements, the digests were separated by reversed phase column chromatography using an EASY nLC 1200 (Thermo-Fisher) with self-packed columns (OD 360 µm, ID 100 µm, length 20 cm) filled with 3 µm diameter C18 particles (Dr. Maisch, Ammerbuch-Entringen, Germany) in a one-column setup. Following loading/desalting in 0.1% acetic acid in water, the peptides were separated by applying a binary non-linear gradient from 5% to 53% acetonitrile in 0.1% acetic acid over 82 min. The LC was coupled online to a LTQ Orbitrap Elite mass spectrometer (Thermo-Fisher) with a spray voltage of 2.5 kV. After a survey scan in the Orbitrap ($r$=60,000), MS/MS data were recorded for the twenty most intensive precursor ions in the linear ion trap. Singly charged ions were not considered for MS/MS analysis. The lock mass option was enabled throughout all analyzes.

After mass spectrometric measurement, database search against a database of *D. discoideum* downloaded from dictyBase (http://dictybase.org/) on 29/04/2022 (12,321 entries) as well as label-free quantification – LFQ (*Cox et al., 2014*) and iBAQ (*Schwanhäusser et al., 2011*) – was performed using MaxQuant (version 1.6.7.0) (*Cox and Mann, 2008*). Common laboratory contaminants and reversed sequences were included by MaxQuant. Search parameters were set as follows: trypsin/P--specific digestion with up to two missed cleavages, methionine oxidation and N-terminal acetylation as variable modification, match between runs with default parameters enabled. The FDRs (false discovery rates) of protein and PSM (peptide spectrum match) levels were set to 0.01. Two identified unique peptides were required for protein identification. LFQ was performed using the following settings: LFQ minimum ratio count 2 considering only unique for quantification.

Results were filtered for proteins quantified in at least two out of three biological replicates before statistical analysis. Here, both strains were compared by a student's t-test applying a threshold p-value of 0.01, which was based on all possible permutations. Proteins were considered to be differentially abundant if the log2LFQ-fold change was greater than |0.8|. The dataset was also filtered for so-called 'on-off' proteins. These proteins are interesting candidates as their changes in abundance might be so drastic that their abundance is below the limit of detection in the 'off' condition. To robustly filter for these proteins, 'on' proteins were defined as being quantified in all three biological replicates of the Δ*sey1* mutant setup and in none of the replicates of the Ax3 wild-type setup, whereas 'off' proteins were quantified in none of the three biological replicates of the mutant setup but in all replicates of the wild-type setup. In addition, the dataset was also filtered for the 50 most abundant proteins in each sample (based on the corresponding iBAQ value). After filtering, we ended with a list of 192 proteins that were either differentially abundant on LDs from *D. discoideum* Ax3 (39 proteins) or Δ*sey1* (76 proteins), or that belonged to the 'on' (7 proteins) / 'off' (22 proteins) categories or to the 50 most abundant proteins (48 additional proteins) (*Supplementary file 1*).

We then functionally mapped these proteins manually according to dictyBase classification, resulting in 135 functionally assigned and 57 unassigned proteins, and generated Voronoi treemaps (Paver v2.1, Decodon) to visualize functional clusters and abundance differences between LDs from *D. discoideum* Ax3 and Δ*sey1*. Voronoi diagrams were chosen as the output, because this visualization allows a functional clustering of proteins and at the same time is more intuitive by correlating protein abundancies to the sizes of the mosaic tiles in the plot. Individual iBAQ values of proteins on

*D. discoideum* Ax3 LDs were represented by the area of the mosaic tiles (large tiles: more abundant, small tiles: less abundant). Comparison of protein abundance on Ax3 LDs with abundance of proteins on Δ*sey1* LDs is represented by the color gradient of the mosaic tiles (orange: more abundant on Ax3 LDs, grey: equally abundant on Ax3 and Δ*sey1* LDs, blue: more abundant on Δ*sey1* LDs).

## Isotopologue profiling

To label *L. pneumophila* intracellularly growing in *D. discoideum* with $^{13}$C-substrates, we used previously published protocols with minor modifications (*Heuner and Eisenreich, 2013*; *Heuner et al., 2019*). In brief, *D. discoideum* Ax3 or Δ*sey1* mutant producing calnexin-GFP (CnxA-GFP) were cultivated in eight T75 cell culture flasks per bacterial strain. After the cells reached confluency (~2×10$^7$ cells per flask), the amoeba were infected with *L. pneumophila* JR32 or Δ*fadL* (MOI 50), by adding bacteria grown in AYE to an OD$_{600}$ of 5 at appropriate dilutions. The flasks were then centrifuged to synchronize infection (500 *g*, 10 min) and incubated for 1 hr at 25 °C. To remove extracellular bacteria, cells were washed once with 10 ml SorC buffer (RT), overlaid newly with 10 ml SorC buffer and further incubated at 25 °C. At 5 hr p.i., 200 μM [U-$^{13}$C$_{16}$]palmitate was added to the flasks, and the cells were further incubated for 10 hr.

At 15 hr p.i., the amoeba were detached using a cell scraper, transferred into 50 ml Falcon tubes, frozen at –80 °C for 1 hr and again thawed to RT. The suspension was then centrifuged at 200 *g* for 10 min at 4 °C, and the supernatant was transferred to new 50 ml Falcon tubes. The pellet, representing eukaryotic cell debris (F1) was washed twice with 50 ml and once with 1 ml cold SorC buffer. The supernatant harvested after the first centrifugation step contained *L. pneumophila* bacteria (F2). This fraction was centrifuged at 3500 *g* for 15 min at 4 °C, and the resulting pellet was washed twice with 50 ml and once with 1 ml cold SorC buffer. The supernatant of F2 was filtered through a 0.22 μm pore filter to remove bacteria, and then 100% trichloroacetic acid (TCA) was added to a final concentration of 10%. The supernatant was incubated on ice for 1 hr and centrifuged at 4600 *g* for 30 min at 4 °C. The resulting pellet (F3) contained cytosolic proteins of *D. discoideum*. The pellets of F1, F2, and F3 were autoclaved at 120 °C for 20 min, freeze-dried and stored at –20 °C until analysis. To monitor cell lysis and the purity of F1 and F2, samples were analyzed by light microscopy.

For isotopologue profiling of amino acids and polyhydroxybutyrate (PHB), bacterial cells (approximately 10$^9$) or 1 mg of the freeze-dried host protein fraction were hydrolysed in 0.5 ml of 6 M HCl for 24 hr at 105 °C, as described earlier (*Eylert et al., 2010*). The HCl was removed under a stream of nitrogen, and the remainder was dissolved in 200 μl acetic acid. The sample was purified on a cation exchange column of Dowex 50Wx8 (H$^+$ form, 200–400 mesh, 5×10 mm), which was washed previously with 1 ml methanol and 1 ml ultrapure water. The column was eluted with 2 ml distilled water (eluate 1) and 1 ml 4 M ammonium hydroxide (eluate 2). An aliquot of the respective eluates was dried under a stream of nitrogen at 70 °C. The dried remainder of eluate 2 was dissolved in 50 μl dry acetonitrile and 50 μl *N*-(tert-butyldimethylsilyl)-*N*-methyltrifluoroacetamide containing 1% tert-butyldimethylsilyl chloride (Sigma) and kept at 70 °C for 30 min. The resulting mixture of tert-butyldimethylsilyl derivates (TBDMS) of amino acids was used for further GC/MS analysis. Due to acid degradation, the amino acids tryptophan and cysteine could not be detected with this method. Furthermore, the hydrolysis condition led to the conversion of glutamine and asparagine to glutamate and aspartate. PHB was hydrolysed to its monomeric component 3-hydroxybutyric acid (3-HB). For derivatization of 3-HB, the dried aliquot of eluate 1 was dissolved in 100 μl *N*-methyl-*N*-(trimethylsilyl)-trifluoroacetamide (Sigma) and incubated in a shaking incubator at 110 rpm (30 min, 40 °C). The resulting trimethylsilyl derivative (TMS) of 3-HB derived from PHB was used for GC/MS analysis without further treatment.

GC/MS-analysis was performed with a QP2010 Plus gas chromatograph/mass spectrometer (Shimadzu) equipped with a fused silica capillary column (Equity TM-5; 30 m×0.25 mm, 0.25 μm film thickness; SUPELCO) and a quadrupole detector working with electron impact ionization at 70 eV. A volume of 0.1–6 μl of the sample was injected in 1:5 split mode at an interface temperature of 260 °C and a helium inlet pressure of 70 kPa. With a sampling rate of 0.5 s, selected ion monitoring was used. Data was collected using LabSolution software (Shimadzu). All samples were measured three times (technical replicates). $^{13}$C-excess values and isotopologue compositions were calculated as described before (*Eylert et al., 2008*) including: (i) determination of the spectrum of unlabeled derivatized metabolites, (ii) determination of mass isotopologue distributions of labeled metabolites and (iii) correction of $^{13}$C-incorporation concerning the heavy isotopologue contributions due to the

natural abundances in the derivatized metabolites. For analysis of amino acids, the column was kept at 150 °C for 3 min and then developed with a temperature gradient of 7 °C min$^{-1}$ to a final temperature of 280 °C that was held for 3 min. The amino acids alanine (6.7 min), aspartate (15.4 min), glutamate (16.8 min) were detected, and isotopologue calculations were performed with m/z [M-57]$^+$ or m/z [M-85]$^+$. For the detection of 3-HB derived from PHB, the column was heated at 70 °C for 3 min and then developed with a first temperature gradient of 10 °C min$^{-1}$ to a final temperature of 150 °C. This was followed by a second temperature gradient of 50 °C min$^{-1}$ to a final temperature of 280 °C, which was held for 3 min. The TMS-derivative of 3-HB, was detected at a retention time of 9.1 min, and isotopologue calculations were performed with m/z [M-15]$^+$.

## Statistical analysis

If not stated otherwise, data analysis and statistics were performed in GraphPad Prism (Version 5.01, GraphPad Software Inc) using two-way ANOVA with Bonferroni post-test. Probability values of less than 0.05, 0.01, and 0.001 were used to show statistically significant differences and are represented with *, **, or ***. The value of 'n' represents the number of analyzed cells, LCVs, LDs or video frames per condition. For comparative proteomics, LFQ values were used for statistical testing of differentially abundant proteins. Empirical Bayes moderated t-tests were applied, as implemented in the R/Bioconductor limma package.

## Acknowledgements

We thank Caroline Barisch and Thierry Soldati for providing the GFP-Plin expression construct and the LDs isolation protocol. We also thank Sebastian Grund for technical support during sample preparation before MS analyses. The Center for Microscopy and Image Analysis of the University of Zürich is acknowledged for maintaining and providing equipment, and ScopeM is acknowledged for instrument access at the ETH Zürich. Work in the group of HH was supported by the Swiss National Science Foundation (SNF; 31003 A_175557, 310030_207826). The lab of MP was supported by the NOMIS foundation.

## Additional information

### Funding

| Funder | Grant reference number | Author |
|---|---|---|
| Schweizerischer Nationalfonds zur Förderung der Wissenschaftlichen Forschung | 31003A_175557 | Hubert Hilbi |
| Schweizerischer Nationalfonds zur Förderung der Wissenschaftlichen Forschung | 310030_207826 | Hubert Hilbi |
| NOMIS Stiftung | | Martin Pilhofer |

The funders had no role in study design, data collection and interpretation, or the decision to submit the work for publication.

### Author contributions

Dario Hüsler, Pia Stauffer, Formal analysis, Investigation, Visualization, Methodology, Writing - original draft; Bernhard Keller, Desirée Böck, Anne Ostrzinski, Bianca Striednig, A Leoni Swart, Formal analysis, Investigation, Methodology, Writing – review and editing; Thomas Steiner, Simone Vormittag, Formal analysis, Investigation, Visualization, Methodology, Writing – review and editing; François Letourneur, Resources, Methodology, Writing – review and editing; Sandra Maaß, Resources, Data curation, Software, Formal analysis, Supervision, Visualization, Methodology, Writing – review and editing; Dörte

Becher, Resources, Supervision, Funding acquisition, Methodology, Writing – review and editing; Wolfgang Eisenreich, Martin Pilhofer, Resources, Formal analysis, Supervision, Funding acquisition, Methodology, Writing – review and editing; Hubert Hilbi, Conceptualization, Resources, Formal analysis, Supervision, Funding acquisition, Writing - original draft, Project administration

#### Author ORCIDs
Pia Stauffer ⓘ http://orcid.org/0009-0008-5742-8362
Bianca Striednig ⓘ http://orcid.org/0000-0001-7575-8965
François Letourneur ⓘ http://orcid.org/0000-0003-2232-6127
Dörte Becher ⓘ http://orcid.org/0000-0002-9630-5735
Hubert Hilbi ⓘ http://orcid.org/0000-0002-5462-9301

#### Decision letter and Author response
Decision letter https://doi.org/10.7554/eLife.85142.sa1
Author response https://doi.org/10.7554/eLife.85142.sa2

## Additional files

#### Supplementary files
• Supplementary file 1. Comparative proteomics analysis of LDs from *D. discoideum* Ax3 or Δ*sey1* mutant amoeba. Dark blue: exclusively present on Δ*sey1* LDs, light blue: more abundant on Δ*sey1* LDs, grey: equally abundant on Ax3 and Δ*sey1* LDs, light orange: more abundant on Ax3 LDs, dark orange: exclusively present on Ax3 LDs.
• Supplementary file 2. Cells, bacterial strains, and plasmids used in this study.
• Supplementary file 3. Oligonucleotides used in this study.
• MDAR checklist

#### Data availability
All data generated or analysed during this study are included in the manuscript and supporting files. The MS proteomics data discussed in this publication have been deposited to the ProteomeXchange Consortium via the PRIDE (*Perez-Riverol et al., 2019*) partner repository with the dataset identifier PXD038200.

The following dataset was generated:

| Author(s) | Year | Dataset title | Dataset URL | Database and Identifier |
| --- | --- | --- | --- | --- |
| Hüsler D, Stauffer P, Keller B, Böck D, Steiner T, Ostrzinski A, Striednig B, Swart AL, Letourneur F, Maaß S, Becher D, Eisenreich W, Pilhofer M, Hilbi H | 2023 | The large GTPase Sey1/atlastin mediates lipid droplet- and FadL-dependent intracellular fatty acid metabolism of *Legionella pneumophila* | https://www.ebi.ac.uk/pride/archive/projects/PXD038200 | PRIDE, PXD038200 |

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
