## [Editor Report]

This important study advances our understanding of host-derived lipid droplets (LDs) interaction with intracellular pathogens. The use of amoeba species *Dictyostelium* discoideum as a host for *Legionella pneumophila* infection is compelling and goes beyond the current state of the art. The data were collected and analyzed using convincing methodology and this paper will interest cell biologists and microbiologists working on the interaction of microbes with host cells.

---

## [Decision Letter]

**Decision letter after peer review:**

Thank you for submitting your article "The large GTPase Sey1/atlastin mediates lipid droplet- and FadL-dependent intracellular fatty acid metabolism of *Legionella pneumophila*" for consideration by *eLife*. Your article has been reviewed by 2 peer reviewers, and the evaluation has been overseen by a Reviewing Editor and Suzanne Pfeffer as the Senior Editor. The reviewers have opted to remain anonymous.

Essential revisions:

The main concerns and essential experiments required are listed below:

1) Perform experiments to understand better how Sey1 or LegG1 promote LD recruitment to LCVs. And provide further demonstrations that LDs indeed enter the LCV and how this uptake occurs.

2) Address concerns about the experiments performed with cells continuously treated with palmitate.

3) Perform additional experiments to confirm that Sey1 localizes to LDs and address the concerns about contaminations in the mass spec data.

4) perform experiments to test the alternative hypothesis that LD localizes only with damaged LCV.

5) In addition, please address the comments related to all technical concerns raised by the reviewers. For example, validate the main differentially expressed proteins by western blot along with Plin as control, among others.

*Reviewer #1 (Recommendations for the authors):*

In general, is it necessary to continuously feed the cells with 200uM palmitate? There are plenty of LDs present even in unfed cells (Figure S1B). I assume it is to increase the number of LDs per cell, but at the same time, it may result in the emergence of artifacts that are not normally observed in "non-obese" cells. Related to this (FigS1), is the effect specific for palmitate, or does the addition of other charged or neutral lipids have a similar effect, which would suggest that any well-fed *Dictyostelium* population might promote *L. pneumophila* growth?

In Figure 1A, the authors show instances where LDs associate with LCVs, and they propose "integration or even dissolution of the limiting membrane". Do they envision this to be the precursor for LD uptake? If so, what is the extent to which LDs show this type of 'dissolution' vs normal association? Related to this and the LD uptake proposed in Figure 5, what do the authors envision this membrane dissolution would be good for? Given that LDs are surrounded only by a lipid monolayer, fusion with the outer leaflet of the LCV would not result in LDs entering the lumen. Instead, they would be sandwiched between two lipid monolayers with nowhere to go. If the authors claim that LDs enter the LCV they need to provide molecular evidence for some sort of uptake mechanism, which I consider the main selling point of the paper.

Searching the literature, I could not find any information about LDs fusing with other cell organelles (except homotypic fusion between LDs). LDs typically engage in "kiss-and-run" or "anchoring' interactions. If no natural fusion machinery exists for LDs, it is highly unlikely that Lp has "invented" one as pathogens usually exploit existing mechanisms /machineries. Again, if uptake really happens then it is imperative to define the mechanism and maybe show the actual uptake process in real time.

Figure 1C, S2: Given that the signals for Sey1, RanBP, and RanA are so abundant within cells, it is difficult to determine what organelles they do or do not colocalize with. It would be more convincing if negative examples were provided (and quantified) for comparison – organelles (maybe mitochondria or lysosomes) that are not expected to colocalize with Sey1. Another potential control would be to tag different ER proteins and see how they compare. Without such reference points, the data shown here for LCVs are difficult to interpret.

Figure 1B: I am unfamiliar with the Voronoi diagram and am not sure what additional information it is giving us over something like a volcano plot or heatmap arranged by the function of the proteins. I also don't understand the distinction of the "black frames" which I also have a difficult time differentiating from the other lines.

More importantly, the authors claim that the mass spec data confirmed that Sey1 localizes to LDs. The data for that are not very convincing. The 'rank abundance' of Sey1 is so low even in the WT sample, and hardly different from the rank abundance of Sey1 in what is purportedly a sey1 KO. I am perhaps missing something or not understanding their mass spec data, but combined with their microscopy data, I am skeptical about their claim that Sey1 is present on LDs. Was it confirmed that the density gradient LDs were pure with nothing contaminating them?

Likewise, in Figure 4 the authors state that the difference in their +/- Sey1 experiments is due to Sey1 being present in LD fractions promoting the process in a GTP-dependent manner. They should probe the LD fractions by western blot to see if (how much) Sey1 is present in LD fraction from wt cells vs KO cells. Also, what happens if LDs are isolated from cells producing Sey1 mutants that constitutively bind either GDP or GTP? This should render the experiment insensitive to GTP addition and address if the stimulatory effect of GTP indeed acts through Sey1 or maybe some other protein that binds GTP.

The authors make use of a sey1 KO cell line. Since Sey1 is of such central importance for ER function, it would be recommended to examine how deleting sey1 globally affects protein levels, not just those related to the LDs. This goes back to the initial criticism noted above regarding how to distinguish direct from indirect effects. Another point to consider is that sey1 deletion might affect the lipid composition of LDs. Is there a difference in fatty acid content between the Ax3 and ∆sey1 LDs that might explain why Lp grows less proficiently?

In various figures, the authors quantify LCV-LD dynamics. It would help if the authors could define exactly what they count as "contact". Second, I personally do not see the utility of the Contact or Mean LD number/LCV metrics. I think all experiments should instead have LD number/ LCV and if the microscopy is live, also include the retention time if desired.

At this point, I would also like to see complementation of the ∆legG1 phenotypes, both with the wild-type protein and known mutant forms that lack certain activities. This goes back to the criticism stated above that the underlying molecular mechanism of their model has not been examined in detail. Also, do LegG1 mutants fail to recruit RanA which would be expected given the reduction in LD contacts? Can deletion of legG1 be rescued by producing constitutively active Ran GTPase? These types of experiments would help to solidify the molecular details of the proposed recruitment mechanism.

Related to this, it is a bit surprising to see that the effects of deleting Sey1 and LegG1 are additive (Figure 2E, F). If LegG1/RanGTPase and Sey1 function in the same pathway as the authors propose, then deletion of either of those genes should lead to a complete loss of LD recruitment.

Since Lpg1810 has not been characterized before, the authors should confirm in synthetic media if FadL indeed mediates lipid uptake by L. pneumophila.

Lastly, the authors should experimentally exclude the possibility that LDs are detected only within LCVs whose integrity has been compromised. It is known from earlier studies that LCVs, especially during isolation from cells, can rupture, which would render them permissive to cell organelles, including LDs. I realize that most of the lumenal LDs lack perilipin, but that could have a reason other than shedding during uptake.

*Reviewer #2 (Recommendations for the authors):*

Specific comments:

1. The involvement of Type 3 and Type 4 secretion system factors has been observed in the induction of lipid remodeling and LD biogenesis in several bacterial infections (PMID: 34783412, PMID: 18333886, PMID: 18486438, PMID: 29390006). Since one of the main objectives of the article is to understand the phagosome/lipid droplets relationship, it is important to add the LD/phagosome dynamic also in control cells not treated with palmitate to figure 1.

2. Still in this context, a high level of fatty acid is very cytotoxic for many cells, please add evidence that this concentration of palmitate does not interfere with the D. discoideum cell viability.

3. Bacterial replication data are one of the bases supporting the authors' hypothesis that LDs favor bacterial replication. I strongly suggest that this data be included in Figure 1.

4. The conditions used for LD purification for proteomic studies were not clearly described in the methods section. Was the purification performed in palmitate-stimulated cells or not?

5. One of the challenges of analyzing the proteome of isolated LDs is the degree of enrichment of that organelle. Due to the limitations of the model, I recommend that authors highlight the bona fide LD-resident proteins found in the LD proteome. Furthermore, it would be important to validate the main differentially expressed proteins by western blot along with Plin as control. Did the authors observe any protein related to the pathogen/host interaction in the proteome of this organelle?

6. The phospholipase PldA is one of the main proteins differentially expressed in D. discoideum Δsey1. It would be highly informative to use phospholipase inhibitors to assess the contribution of phospholipases to L. pneumophila LD utilization and lipid metabolism.

7. Another control that needs to be included in the manuscript is 13C-Excess (mol%) in key metabolites of amoeba. This control is important for distinguishing the metabolites that are the result of L. pneumophila metabolism and those that are obtained from the host cell.

8. Some of the legends do not include the number of replicates and n of samples used. The information should be included in all experiments.

9. In Figure 6, the fold change of bacterial replication should be included. Still in this context, it is important to include in the discussion that the LDs and lipid metabolism seem to be important for optimal bacterial replication, but not essential for L. pneumophila replication and infection.

---

## [Author Response]

Essential revisions:The main concerns and essential experiments required are listed below:1) Perform experiments to understand better how Sey1 or LegG1 promote LD recruitment to LCVs. And provide further demonstrations that LDs indeed enter the LCV and how this uptake occurs.

We performed additional experiments to address the role of the *L. pneumophila* effector LegG1 for LD-LCV interactions. To this end, (i) we visualized microtubules (GFP-tubulin A) in the *D. discoideum* parental strain Ax3 and Δ*sey1* mutant amoeba infected with *L. pneumophila* JR32, Δ*icmT*, Δ*legG1* or Δ*legG1*/pLegG1 (new Figure 4D), (ii) we assessed the subcellular localization of GFP-LegG1 (new Figure 4—figure supplement 2), and (iii) we complemented the LD-LCV interaction phenotype of the Δ*legG1* mutant strain by providing the *legG1* gene on a plasmid (new Figure 4C).

Regarding the entry of LDs into the LCV lumen, we now extensively outline in the Discussion section mechanistic aspects possibly underlying the LCV-LD interactions (l. 490-515). We now also outline in the Discussion section studies showing that LDs also enter the *Mycobacterium marinum*-containing vacuole in *D. discoideum*, and the lipids are stored in the bacteria as “intracytosolic lipid inclusions” (ILI) (Barisch et al., 2015, Barisch et al., 2017, Barisch & Soldati, 2017) (l. 449-453). Moreover, we discuss the LD uptake into *Chlamydia* inclusions, which proceeds through the formation of an effector protein-mediated LD-inclusion contact site, followed by what appears to be an endocytic uptake leading to membrane-coated, intra-luminal LDs (Cocciaro et al., 2008). We did not observe such structures of LDs in LCVs (l. 454-468).

Finally, regarding the role(s) of FadL for palmitate-mediated effects on *L. pneumophila*, we tested whether palmitate affects the growth of *L. pneumophila* wild-type or Δ*fadL* (new Figure 7—figure supplement 2).

2) Address concerns about the experiments performed with cells continuously treated with palmitate.

We performed a number of additional experiments to address the question, whether the LCV-LD interactions are quantitatively and/or qualitatively affected by feeding *D. discoideum* with palmitate (new Figure 1—figure supplement 1, Figure 3—figure supplement 1). Overall, these experiments revealed that feeding with palmitate increased the number but not the size of LDs in *D. discoideum*, and infection with wild-type *L. pneumophila* reduced the LDs number compared to uninfected or ∆*icmT*-infected amoeba. However, in apparent contrast to mammalian cells, Sey1 did not seem to affect the number and size of LDs in *D. discoideum*. Overall, feeding *D. discoideum* with palmitate affected the LCV-LD interactions quantitatively, but not qualitatively.

3) Perform additional experiments to confirm that Sey1 localizes to LDs and address the concerns about contaminations in the mass spec data.

In order to assess the localization of Sey1 in *D. discoideum*, we constructed and used GFP-Sey1 and Sey1-GFP probes. We now show the subcellular localization of GFP-Sey1 in intact *D. discoideum* as well as in amoeba homogenates. The highly produced GFP fusion proteins localized in host cells in a punctate manner but rather ambiguously. Unfortunately, no anti-Sey1 antibodies are available for *D. discoideum*, and therefore, immune-fluorescence localization experiments cannot be performed. Finally, to document the purity of the LD preparation, we now show an overview of the purification steps, leading to a highly enriched LD preparation (new Figure 4—figure supplement 1).

4) Perform experiments to test the alternative hypothesis that LD localizes only with damaged LCV.

We assessed LCV integrity during *L. pneumophila* infection of *D. discoideum* Ax3 and Δ*sey1* mutant amoeba producing P4C-GFP and cytoplasmic mCherry (new Figure 2DE). LCVs are impermeable to cytoplasmically produced mCherry, and therefore, the fluorescent protein is a valid tool to assess LCV integrity (Koliwer-Brandl et al., 2019). This approach revealed that 1 h post infection, all LCVs formed in *D. discoideum* Ax3 or Δ*sey1* were impermeable to cytoplasmic mCherry, and therefore, LDs translocate into LCVs without compromised membrane integrity.

5) In addition, please address the comments related to all technical concerns raised by the reviewers. For example, validate the main differentially expressed proteins by western blot along with Plin as control, among others.

We addressed all the technical concerns raised by the reviewers. Unfortunately, Western blot and immune-fluorescence controls are not possible due to lack of suitable antibodies for *D. discoideum*. Please find below a point-by-point response to the reviewers’ concerns.

Reviewer #1 (Recommendations for the authors):In general, is it necessary to continuously feed the cells with 200uM palmitate? There are plenty of LDs present even in unfed cells (Figure S1B). I assume it is to increase the number of LDs per cell, but at the same time, it may result in the emergence of artifacts that are not normally observed in "non-obese" cells. Related to this (FigS1), is the effect specific for palmitate, or does the addition of other charged or neutral lipids have a similar effect, which would suggest that any well-fed *Dictyostelium* population might promote *L. pneumophila* growth?

Thank you for raising this important point. We performed additional experiments to quantify and compare the number of LDs per cell and their size in unfed and in palmitate-fed amoeba (new Figure 4—figure supplement 1). Uninfected *D. discoideum* strain Ax3 and ∆*sey1* mutant amoeba and the strains infected with either *L. pneumophila* wild-type or ∆*icmT* harbored similar numbers of LDs. Upon feeding the amoeba with 200 µM palmitate, all amoeba contained more LDs, and the infection with wild-type *L. pneumophila* reduced the number of LDs compared to uninfected, palmitate-fed amoeba (new Figure 4—figure supplement 1). These results are now outlined in the text (l. 205-229).

We also performed additional experiments to quantify the ratio of LDs per LCV and LDs per LCV area in unstimulated *D. discoideum* Ax3 or ∆*sey1* mutant amoeba (new Figure 4—figure supplement 1). In unfed *D. discoideum*, the number of LDs per LCV and the number of LDs per LCV area were significantly larger in the parental *D. discoideum* strain Ax2 as compared to Δ*sey1* mutant amoeba. These findings are similar to amoeba stimulated with 200 µM palmitate. However, the overall number of LDs per LCV or LDs per LCV area increased upon feeding the amoeba with palmitate in agreement with an increased overall number of LDs per cell. Taken together, these results indicate that while feeding *D. discoideum* with palmitate increases the total number of LDs, it does not change the positive effect of Sey1 on the ratio of LDs per LCV or LCV area. Accordingly, our findings about LD-LCV interactions in palmitate-fed *D. discoideum* are also valid for unstimulated amoeba. These results are now outlined in the text (l. 219-229).

In Figure 1A, the authors show instances where LDs associate with LCVs, and they propose "integration or even dissolution of the limiting membrane". Do they envision this to be the precursor for LD uptake? If so, what is the extent to which LDs show this type of 'dissolution' vs normal association? Related to this and the LD uptake proposed in Figure 5, what do the authors envision this membrane dissolution would be good for? Given that LDs are surrounded only by a lipid monolayer, fusion with the outer leaflet of the LCV would not result in LDs entering the lumen. Instead, they would be sandwiched between two lipid monolayers with nowhere to go. If the authors claim that LDs enter the LCV they need to provide molecular evidence for some sort of uptake mechanism, which I consider the main selling point of the paper.

The cryo-tomograms in Figure 1B showing intimate interactions of LDs with the limiting LCV membrane in *D. discoideum* Ax3 infected with *L. pneumophila* JR32 are representative, and we did not observe “normal” associations, i.e., structures that look like membrane contact sites. We now explicitly state this finding in the text (l. 163-165). LDs entering a pathogen vacuole have been previously described for *Mycobacterium marinum* in *D. discoideum* and for *Chlamydia trachomatis* in HeLa cells. In these studies, the authors show that LDs translocate from the host cytoplasm to the pathogen vacuole lumen. In the case of *C. trachomatis*, the authors show that LDs enter the pathogen vacuole (inclusion) lumen by invagination of the inclusion membrane, resulting in membrane-decorated intra-luminal LDs. We did not observe such structures of LDs in LCVs. We now outline these results in the Discussion section (l. 466-468) and further discuss the implications of the LD monolayer interacting with the LCV bilayer (l. 435-441). The mechanism of LD translocation of the LCV membrane is certainly an interesting topic, which will be addressed in future studies.

Searching the literature, I could not find any information about LDs fusing with other cell organelles (except homotypic fusion between LDs). LDs typically engage in "kiss-and-run" or "anchoring' interactions. If no natural fusion machinery exists for LDs, it is highly unlikely that Lp has "invented" one as pathogens usually exploit existing mechanisms /machineries. Again, if uptake really happens then it is imperative to define the mechanism and maybe show the actual uptake process in real time.

Thank you for highlighting these very interesting mechanistic aspects. While LDs indeed do not seem to fuse with other organelles, they are formed within and released from the ER bilayer. The release is reversible, and accordingly, cytosolic LDs interact with and re-integrate into the ER (Thiam *et al.*, 2013, Wilfling *et al.*, 2013, Wilfling *et al.*, 2014). The process is dependent on the small GTPase Arf1 and the coatomer complex-I (COPI), but otherwise is ill defined, and specific “integration factors” have not been identified. Analogously, LDs might interact with and integrate into the LCV limiting membrane. We have now outlined these thoughts in the Discussion section (l. 435-441).

This raises the intriguing possibility that *L. pneumophila* might indeed have “invented” an effector protein promoting the LD-LCV integration. We agree with the reviewer that pathogens usually exploit existing mechanisms/machineries. However, *L. pneumophila* also produces effectors, which catalyze seemingly novel cell biological reactions. A prominent example is the SidE family of ubiquitin ligases, which catalyze ubiquitination through a single effector protein showing ADP-ribosylation and nucleotidase/ phosphohydrolase activity thus bypassing the canonical E1-E2-E3 ubiquitin ligase system.

Figure 1C, S2: Given that the signals for Sey1, RanBP, and RanA are so abundant within cells, it is difficult to determine what organelles they do or do not colocalize with. It would be more convincing if negative examples were provided (and quantified) for comparison – organelles (maybe mitochondria or lysosomes) that are not expected to colocalize with Sey1. Another potential control would be to tag different ER proteins and see how they compare. Without such reference points, the data shown here for LCVs are difficult to interpret.

It has been published that in *D. discoideum* GFP-Sey1 localizes to the ER (Steiner *et al.*, 2017), and RanA-GFP as well as RanBP1-GFP localize to LCVs (Rothmeier *et al.*, 2013). In this study, we aimed at validating the localization of these GFP fusion proteins to LDs. We agree with the reviewer that the abundance of the GFP fusion proteins in *D. discoideum* renders it difficult to judge whether they actually co-localize with LDs or not. However, given the abundance of cell organelles like mitochondria or lysosomes, co-localization studies with these organelles would likely also not allow to determine a more precise subcellular localization of the GFP fusion proteins. For a better comparison among each other, we now show all the localization experiments in one figure, new Figure 4—figure supplement 1. Moreover, we now also assess the localization of GFP-LegG1 with regard to LDs in *D. discoideum* Ax3 and ∆*sey1* (new Figure 4—figure supplement 2), and we analyze the effect of LegG1 on microtubule formation in *D. discoideum* using GFP-tubulin A (new Figure 4D). Intriguingly, the *L. pneumophila* ∆*legG1* mutant strain impairs microtubule stability in *D. discoideum* Ax3, as published in Rothmeier et al., 2013, as well as in the ∆*sey1* mutant strain. The latter experiment contributes to elucidating the role of LegG1 for LCV-LD interactions, since both organelles move along microtubule filaments within the cell.

Figure 1B: I am unfamiliar with the Voronoi diagram and am not sure what additional information it is giving us over something like a volcano plot or heatmap arranged by the function of the proteins. I also don't understand the distinction of the "black frames" which I also have a difficult time differentiating from the other lines.

The volcano plot is classically used for plotting fold-changes of protein abundancies versus p-values without allowing a representation of functional assignment of proteins, and no representation of the absolute protein amounts is possible. In heatmap plots a functional clustering of proteins is possible (as noted by the reviewer), but the absolute protein amounts can only be represented as color codes. We believe that the Voronoi diagram partially overcomes these shortcomings, as it allows a functional clustering of proteins and at the same time is more intuitive by correlating protein abundancies to the sizes of the different mosaic tiles in the plot. These reflections are now outlined in the Materials and methods section (l. 832-834).

More importantly, the authors claim that the mass spec data confirmed that Sey1 localizes to LDs. The data for that are not very convincing. The 'rank abundance' of Sey1 is so low even in the WT sample, and hardly different from the rank abundance of Sey1 in what is purportedly a sey1 KO. I am perhaps missing something or not understanding their mass spec data, but combined with their microscopy data, I am skeptical about their claim that Sey1 is present on LDs. Was it confirmed that the density gradient LDs were pure with nothing contaminating them?

In mammalian cells, the homologs of Sey1, atlastin 1-3 localize to LDs and determine their size and number (Klemm et al., 2013). We agree with the reviewer that based on our MS and fluorescence microscopy experiments the evidence that Sey1 localizes to LDs in *D. discoideum* is not very strong, and Sey1 might affect the LCV-LD dynamics in a more indirect manner. Unfortunately, there are no antibodies available, which recognize *D. discoideum* Sey1. Taking into consideration these ambiguities, we state in the manuscript (l. 253-256): “Under the conditions used, GFP-Sey1 accumulated in the vicinity of LDs in intact cells as well as in cell homogenates, but apparently did not co-localize with LDs. This staining pattern suggests that Sey1 localizes only in very low amounts to LDs, or that localization of Sey1 to LDs is impaired due to the fluorescent protein tag.” Moreover, we now document in the manuscript the enrichment of mCherry-Plin-positive LDs upon purification by sucrose density gradient centrifugation (new Figure 4—figure supplement 1, l. 233-236).

Likewise, in Figure 4 the authors state that the difference in their +/- Sey1 experiments is due to Sey1 being present in LD fractions promoting the process in a GTP-dependent manner. They should probe the LD fractions by western blot to see if (how much) Sey1 is present in LD fraction from wt cells vs KO cells. Also, what happens if LDs are isolated from cells producing Sey1 mutants that constitutively bind either GDP or GTP? This should render the experiment insensitive to GTP addition and address if the stimulatory effect of GTP indeed acts through Sey1 or maybe some other protein that binds GTP.

Given the ambiguity of the results, we down-tone the statement that Sey1 localizing to LDs in *D. discoideum* is directly causing the LCV-LD interactions (see above, point 6), text (l. 253-254) and Discussion section (l. 501-515). As stated above, antibodies recognizing *D. discoideum* Sey1 are unfortunately not available. Moreover, *D. discoideum* Sey1 mutants that constitutively bind GDP or GTP are also not available and also would likely not allow to distinguish between direct effects of Sey1 on LDs versus more indirect effects on, e.g., ER-dependent LD biogenesis.

The authors make use of a sey1 KO cell line. Since Sey1 is of such central importance for ER function, it would be recommended to examine how deleting sey1 globally affects protein levels, not just those related to the LDs. This goes back to the initial criticism noted above regarding how to distinguish direct from indirect effects. Another point to consider is that sey1 deletion might affect the lipid composition of LDs. Is there a difference in fatty acid content between the Ax3 and ∆sey1 LDs that might explain why Lp grows less proficiently?

We thoroughly characterized the pleiotropic phenotypes of *D. discoideum* ∆*sey1* mutant amoeba in a previous publication (Hüsler et al., 2021). *D. discoideum* lacking *sey1* shows an altered ER architecture, and the tubular ER network is partially disrupted without any major consequences for other organelles or the architecture of the secretory and endocytic pathways. In agreement with the many cellular roles of the ER, in the ∆*sey1* mutant amoeba, macropinocytic and phagocytic functions are preserved, but lysosomal enzymes exocytosis, intracellular proteolysis, cell motility, growth on bacterial lawns and LCV expansion as well as intracellular replication of *L. pneumophila* are impaired. Accordingly, Sey1 affects protein levels in a plethora of cellular pathways. These results are now stated in the manuscript (l. 513-521).

Regarding the fatty acid composition of LDs from *D. discoideum* Ax3 and ∆*sey1*: the two *Dictyostelium* strain harbor LDs of similar numbers per cell and of similar size (Figure 3—figure supplement 1). Hence, given that Sey1 does not seem to affect the overall number size and number of LDs, it seems unlikely that the fatty acid content of LDs is substantially different in the *D. discoideum* parental strain Ax3 versus ∆*sey1* mutant amoeba. We now outline these reflections in the Discussion section (l. 501-506).

In various figures, the authors quantify LCV-LD dynamics. It would help if the authors could define exactly what they count as "contact". Second, I personally do not see the utility of the Contact or Mean LD number/LCV metrics. I think all experiments should instead have LD number/ LCV and if the microscopy is live, also include the retention time if desired.

We now define “contact” in the Material and Methods section (l. 726-728): “An LCV-LD “contact” was defined as the visual association of the signal on the LCV-limiting membrane (P4C-GFP, AmtA-GFP) with the LD mCherry-Plin signal; and only Plin/LipidTOX Deep Red-positive LDs were considered for LCV-LD contacts.” As suggested by the reviewer, we changed the figure labels to “LD number / LCV” throughout the manuscript.

At this point, I would also like to see complementation of the ∆legG1 phenotypes, both with the wild-type protein and known mutant forms that lack certain activities. This goes back to the criticism stated above that the underlying molecular mechanism of their model has not been examined in detail. Also, do LegG1 mutants fail to recruit RanA which would be expected given the reduction in LD contacts? Can deletion of legG1 be rescued by producing constitutively active Ran GTPase? These types of experiments would help to solidify the molecular details of the proposed recruitment mechanism.

As suggested by the reviewer, we repeated the experiment addressing the role of LegG1 on the LCV-LD interaction dynamics. The phenotype of the *L. pneumophila* ∆*legG1* mutant strain was complemented by providing the legG1 gene on a plasmid (new Figure 4C), text: l. 279-281. Moreover, we also complemented the microtubule stabilization phenotype of the ∆*legG1* mutant strain (new Figure 4D), text: l. 292-304.

We have previously shown that the *L. pneumophila* ∆*legG1* mutant strain fails to recruit RanA and its effector RanBP1 to LCVs (Rothmeier et al., 2013). This activity might indeed (at least partially) account for the reduced LCV-LD interactions. These reflections are outlined in the Discussion section (l. 414-425). Constitutively active Ran GTPase is unfortunately not available for *D. discoideum*.

Related to this, it is a bit surprising to see that the effects of deleting Sey1 and LegG1 are additive (Figure 2E, F). If LegG1/RanGTPase and Sey1 function in the same pathway as the authors propose, then deletion of either of those genes should lead to a complete loss of LD recruitment.

We do not claim that LegG1/RanA GTPase and Sey1 function in the same pathway. Rather, LegG1/RanA GTPase stabilize microtubules (Rothmeier et al., 2013), and Sey1 promotes homotypic ER tubule fusion and determines the ER architecture (Hüsler et al., 2021). Accordingly, the additive effects of deleting Sey1 and LegG1 might be expected (Figure 4B). These reflections are now outlined in the Results section (l. 292-304).

Since Lpg1810 has not been characterized before, the authors should confirm in synthetic media if FadL indeed mediates lipid uptake by L. pneumophila.

We tested whether the addition of palmitate affects the growth of *L. pneumophila* wild-type or ∆*fadL* mutant bacteria in broth (new Figure 7—figure supplement 2). However, under the conditions used, up to 200 µM palmitate did not affect the growth of these two *L. pneumophila* strains (l. 365-366).

Lastly, the authors should experimentally exclude the possibility that LDs are detected only within LCVs whose integrity has been compromised. It is known from earlier studies that LCVs, especially during isolation from cells, can rupture, which would render them permissive to cell organelles, including LDs. I realize that most of the lumenal LDs lack perilipin, but that could have a reason other than shedding during uptake.

As suggested by the reviewer, we assessed the integrity of the LCVs during their interaction with LDs. To this end, we used *D. discoideum* Ax3 or Δ*sey1* producing P4C-GFP and cytoplasmic mCherry (new Figure 2DE). This approach revealed that 1 h post infection, all LCVs formed in both *D. discoideum* strains were impermeable to cytoplasmic mCherry, and therefore, LDs translocate into LCVs without compromised membrane integrity. This experiment is documented in the new Figure 2DE and outlined in the text (l. 182-191).

Reviewer #2 (Recommendations for the authors):Specific comments:1. The involvement of Type 3 and Type 4 secretion system factors has been observed in the induction of lipid remodeling and LD biogenesis in several bacterial infections (PMID: 34783412, PMID: 18333886, PMID: 18486438, PMID: 29390006). Since one of the main objectives of the article is to understand the phagosome/lipid droplets relationship, it is important to add the LD/phagosome dynamic also in control cells not treated with palmitate to figure 1.

Thank you for raising this important point (see also Reviewer #1, point 1). We have now included in the study a number of control experiments (new Figure 1—figure supplement 1, new Figure 3—figure supplement 1). We performed additional experiments to quantify and compare the number of LDs per cell and their size in unfed and in palmitate-fed amoeba, which were left uninfected or were infected with *L. pneumophila* wild-type or ∆*icmT* (new Figure 3—figure supplement 1). These results are now outlined in the text (l. 209-218). We also performed additional experiments to quantify the ratio of LDs per LCV and LDs per LCV area in unstimulated *D. discoideum* Ax3 or ∆*sey1* mutant amoeba (new Figure 3—figure supplement 1). Overall, these results indicate that feeding *D. discoideum* with palmitate increases the total number of LDs, but it does not change the positive effect of Sey1 on the ratio of LDs per LCV or LCV area. Accordingly, our findings about LD-LCV interactions in palmitate-fed *D. discoideum* are also valid for unstimulated amoeba. These results are now outlined in the text (l. 219-229).

2. Still in this context, a high level of fatty acid is very cytotoxic for many cells, please add evidence that this concentration of palmitate does not interfere with the D. discoideum cell viability.

As suggested, we assessed toxicity of palmitate for *D. discoideum* using Zombie Aqua dye and flow cytometry. Under the conditions used, there was no significant toxicity up to 800 µM palmitate (new Figure 1—figure supplement 1), text: l. 154-156.

3. Bacterial replication data are one of the bases supporting the authors' hypothesis that LDs favor bacterial replication. I strongly suggest that this data be included in Figure 1.

As suggested, the replication data has now been included in Figure 1 (Figure 1A).

4. The conditions used for LD purification for proteomic studies were not clearly described in the methods section. Was the purification performed in palmitate-stimulated cells or not?

We used palmitate-stimulated *D. discoideum* to purify LDs. This is now indicated in the text (l. 233-236) and the Materials and methods section (l. 782-784).

5. One of the challenges of analyzing the proteome of isolated LDs is the degree of enrichment of that organelle. Due to the limitations of the model, I recommend that authors highlight the bona fide LD-resident proteins found in the LD proteome. Furthermore, it would be important to validate the main differentially expressed proteins by western blot along with Plin as control. Did the authors observe any protein related to the pathogen/host interaction in the proteome of this organelle?

In Supplementary File 1, we now highlight in bold bona-fide LD-resident proteins, such as perilipin (Plin, PlnA). Moreover, we now also document the purification scheme and the enrichment of LDs (new Figure 4—figure supplement 1). Interestingly, we identified the small GTPase RanA and its effector RanBP1 in the LD proteome. This unexpected finding spurred the hypothesis that the *L. pneumophila* Ran activator LegG1 might promote LCV-LD interactions. We now highlight in the text that Ran/RanBP1 were unexpectedly identified in the LD proteome (l. 245-248), and the intriguing results are documented in Figure 4 (new Figure 4CD). Unfortunately, there are no antibodies available for *D. discoideum*, which would allow to validate by Western blot the differential production of proteins in strain Ax3 versus ∆*sey1* mutant amoeba.

6. The phospholipase PldA is one of the main proteins differentially expressed in D. discoideum Δsey1. It would be highly informative to use phospholipase inhibitors to assess the contribution of phospholipases to L. pneumophila LD utilization and lipid metabolism.

Given that the topics covered in the study are already quite broad (LDs, Sey1, RanA/RanBP1, LegG1, FadL), we feel that an analysis of the role of the PldA phospholipase for LD-dependent intracellular growth of *L. pneumophila* is beyond the scope of the current study. The phospholipase PldA is certainly an interesting topic, which will be addressed in future studies.

7. Another control that needs to be included in the manuscript is 13C-Excess (mol%) in key metabolites of amoeba. This control is important for distinguishing the metabolites that are the result of L. pneumophila metabolism and those that are obtained from the host cell.

For the isotopologue profiling experiments, we generated 3 fractions: fraction 1 (pellet of low-speed centrifugation of infected amoeba = host cell debris), fraction 2 (pellet of high-speed centrifugation = bacteria) and fraction 3 (precipitated supernatant of high-speed centrifugation = cytoplasmic proteins). While the incorporation of label into amino acids was comparable for fraction 1-3, the incorporation of label into 3-HB is specific for bacterial anabolism, since 3-HB is not produced by eukaryotic cells. This information is now outlined in the text (l. 387-393).

8. Some of the legends do not include the number of replicates and n of samples used. The information should be included in all experiments.

We have now indicated in all figure legends the number of replicates and n of samples used.

9. In Figure 6, the fold change of bacterial replication should be included. Still in this context, it is important to include in the discussion that the LDs and lipid metabolism seem to be important for optimal bacterial replication, but not essential for L. pneumophila replication and infection.

The bacterial growth rate is determined in the exponential phase. The initial infection of *D. discoideum* with *L. pneumophila* is synchronized by centrifugation. The following infection rounds lasting several days are no longer synchronized, and therefore, it is difficult to determine whether and which portion of the intracellular bacteria grow exponentially. Consequently, the fold change of bacterial replication cannot be precisely determined. We now outline in the Discussion section that lipid metabolism and LDs are important but not essential for intracellular replication of *L. pneumophila* (l. 487-489).